# On the Inductive Bias of Neural Tangent Kernels

**Alberto Bietti**
Inria[*]
alberto.bietti@inria.fr

**Julien Mairal**
Inria[*]
julien.mairal@inria.fr

## Abstract

State-of-the-art neural networks are heavily over-parameterized, making the optimization algorithm a crucial ingredient for learning predictive models with good generalization properties. A recent line of work has shown that in a certain over-parameterized regime, the learning dynamics of gradient descent are governed by a certain kernel obtained at initialization, called the *neural tangent kernel*. We study the inductive bias of learning in such a regime by analyzing this kernel and the corresponding function space (RKHS). In particular, we study smoothness, approximation, and stability properties of functions with finite norm, including stability to image deformations in the case of convolutional networks, and compare to other known kernels for similar architectures.

## 1 Introduction

The large number of parameters in state-of-the-art deep neural networks makes them very expressive, with the ability to approximate large classes of functions [26, 41]. Since many networks can potentially fit a given dataset, the optimization method, typically a variant of gradient descent, plays a crucial role in selecting a model that generalizes well [39].

A recent line of work [2, 16, 20, 21, 27, 30, 54] has shown that when training deep networks in a certain over-parameterized regime, the dynamics of gradient descent behave like those of a linear model on (non-linear) features determined at initialization. In the over-parameterization limit, these features correspond to a kernel known as the *neural tangent kernel*. In particular, in the case of a regression loss, the obtained model behaves similarly to a minimum norm kernel least squares solution, suggesting that this kernel may play a key role in determining the inductive bias of the learning procedure and its generalization properties. While it is still not clear if this regime is at play in state-of-the-art deep networks, there is some evidence that this phenomenon of "lazy training" [16], where weights only move very slightly during training, may be relevant for early stages of training and for the outmost layers of deep networks [29, 53], motivating a better understanding of its properties.

In this paper, we study the inductive bias of this regime by studying properties of functions in the space associated with the neural tangent kernel for a given architecture (that is, the reproducing kernel Hilbert space, or RKHS). Such kernels can be defined recursively using certain choices of dot-product kernels at each layer that depend on the activation function. For the convolutional case with rectified linear unit (ReLU) activations and arbitrary patches and linear pooling operations, we show that the NTK can be expressed through kernel feature maps defined in a tree-structured hierarchy.

We study smoothness and stability properties of the kernel mapping for two-layer networks and CNNs, which control the variations of functions in the RKHS. In particular, a useful inductive bias when dealing with natural signals such as images is stability of the output to deformations of the input, such as translations or small rotations. A precise notion of stability to deformations was proposed by Mallat [35], and was later studied in [11] in the context of CNN architectures, showing

---

[*]Univ. Grenoble Alpes, Inria, CNRS, Grenoble INP, LJK, 38000 Grenoble, France

the benefits of different architectural choices such as small patch sizes. In contrast to the kernels studied in [11], which for instance cover the limiting kernels that arise from training only the last layer of a ReLU CNN, we find that the obtained NTK kernel mappings for the ReLU activation lack a desired Lipschitz property which is needed for stability to deformations in the sense of [11, 12, 35]. Instead, we show that a weaker smoothness property similar to Hölder smoothness holds, and this allows us to show that the kernel mapping is stable to deformations, albeit with a different guarantee.

In order to balance our observations on smoothness, we also consider approximation properties for the NTK of two-layer ReLU networks, by characterizing the RKHS using a Mercer decomposition of the kernel in the basis of spherical harmonics [6, 46, 47]. In particular, we study the decay of eigenvalues for this decomposition, which is then related to the regularity of functions in the space, and provides rates of approximation for Lipschitz functions [6]. We find that the full NTK has better approximation properties compared to other function classes typically defined for ReLU activations [6, 17, 19], which arise for instance when only training the weights in the last layer, or when considering Gaussian process limits of ReLU networks (*e.g.*, [24, 28, 36, 40]).

**Contributions.**   Our main contributions can be summarized as follows:

- We provide a derivation of the NTK for convolutional networks with generic linear operators for patch extraction and pooling, and express the corresponding kernel feature map hierarchically using these operators.
- We study smoothness properties of the kernel mapping for ReLU networks, showing that it is not Lipschitz but satisfies a weaker Hölder smoothness property. For CNNs, we then provide a guarantee on deformation stability.
- We characterize the RKHS of the NTK for two-layer ReLU networks by providing a spectral decomposition of the kernel and studying its spectral decay. This leads to improved approximation properties compared to other function classes based on ReLU.

**Related work.**   Neural tangent kernels were introduced in [27], and similar ideas were used to obtain more quantitative guarantees on the global convergence of gradient descent for over-parameterized neural networks [2, 3, 16, 20, 21, 30, 50, 54]. The papers [3, 20, 51] also derive NTKs for convolutional networks, but focus on simpler architectures. Kernel methods for deep neural networks were studied for instance in [17, 19, 34]. Stability to deformations was originally introduced in the context of the scattering representation [12, 35], and later extended to neural networks through kernel methods in [11]. The inductive bias of optimization in neural network learning was considered, *e.g.*, by [1, 4, 13, 39, 48]. [6, 25, 45, 49] study function spaces corresponding to two-layer ReLU networks. In particular, [25] also analyzes properties of the NTK, but studies a specific high-dimensional limit for generic activations, while we focus on ReLU networks, studying the corresponding eigenvalue decays in finite dimension.

## 2   Neural Tangent Kernels

In this section, we provide some background on "lazy training" and neural tangent kernels (NTKs), and introduce the kernels that we study in this paper. In particular, we derive the NTK for generic convolutional architectures on $\ell^2$ signals. For simplicity of exposition, we consider scalar-valued functions, noting that the kernels may be extended to the vector-valued case, as done, *e.g.*, in [27].

### 2.1   Lazy training and neural tangent kernels

Multiple recent works studying global convergence of gradient descent in neural networks (*e.g.*, [2, 20, 21, 27, 30, 54]) show that when a network is sufficiently over-parameterized, weights remain close to initialization during training. The model is then well approximated by its linearization around initialization. For a neural network $f(x; \theta)$ with parameters $\theta$ and initialization $\theta_0$, we then have:[2]

$$f(x; \theta) \approx f(x; \theta_0) + \langle \theta - \theta_0, \nabla_\theta f(x; \theta_0) \rangle. \tag{1}$$

This regime where weights barely move has also been referred to as "lazy training" [16], in contrast to other situations such as the "mean-field" regime (*e.g.*, [15, 38, 37]), where weights move according

to non-linear dynamics. Yet, with sufficient over-parameterization, the (non-linear) features $x \mapsto \nabla_\theta f(x; \theta_0)$ of the linearized model (1) become expressive enough to be able to perfectly fit the training data, by approximating a kernel method.

**Neural Tangent Kernel (NTK).** When the width of the network tends to infinity, assuming an appropriate initialization on weights, the features of the linearized model tend to a limiting kernel $K$, called *neural tangent kernel* [27]:

$$\langle \nabla_\theta f(x; \theta_0), \nabla_\theta f(x', \theta_0) \rangle \rightarrow K(x, x'). \tag{2}$$

In this limit and under some assumptions, one can show that the weights move very slightly and the kernel remains fixed during training [27], and that gradient descent will then lead to the minimum norm kernel least-squares fit of the training set in the case of the $\ell_2$ loss (see [27] and [37, Section H.7]). Similar interpolating solutions have been found to perform well for generalization, both in practice [10] and in theory [8, 31]. When the number of neurons is large but finite, one can often show that the kernel only deviates slightly from the limiting NTK, at initialization and throughout training, thus allowing convergence as long as the initial kernel matrix is non-degenerate [3, 16, 20, 21].

**NTK for two-layer ReLU networks.** Consider a two layer network of the form $f(x; \theta) = \sqrt{\frac{2}{m}} \sum_{j=1}^{m} v_j \sigma(w_j^\top x)$, where $\sigma(u) = (u)_+ = \max(0, u)$ is the ReLU activation, $x \in \mathbb{R}^p$, and $\theta = (w_1^\top, \ldots, w_m^\top, v^\top)$ are parameters with values initialized as $\mathcal{N}(0, 1)$. Practitioners often include the factor $\sqrt{2/m}$ in the variance of the initialization of $v_j$, but we treat it as a scaling factor following [20, 21, 27], noting that this leads to the same predictions. The factor 2 is simply a normalization constant specific to the ReLU activation and commonly used by practitioners, which avoids vanishing or exploding behavior for deep networks. The corresponding NTK is then given by [16, 21]:

$$K(x, x') = 2(x^\top x') \, \mathbb{E}_{w \sim \mathcal{N}(0, I)}[\mathbb{1}\{w^\top x \geq 0\} \mathbb{1}\{w^\top x' \geq 0\}] + 2 \, \mathbb{E}_{w \sim \mathcal{N}(0, I)}[(w^\top x)_+ (w^\top x')_+]$$

$$= \|x\| \|x'\| \kappa \left( \frac{\langle x, x' \rangle}{\|x\| \|x'\|} \right), \tag{3}$$

where

$$\kappa(u) := u \kappa_0(u) + \kappa_1(u) \tag{4}$$

$$\kappa_0(u) = \frac{1}{\pi} \left( \pi - \arccos(u) \right), \qquad \kappa_1(u) = \frac{1}{\pi} \left( u \cdot (\pi - \arccos(u)) + \sqrt{1 - u^2} \right). \tag{5}$$

The expressions for $\kappa_0$ and $\kappa_1$ follow from standard calculations for arc-cosine kernels of degree 0 and 1 (see [17]). Note that in this two-layer case, the non-linear features obtained for finite neurons correspond to a random features kernel [42], which is known to approximate the full kernel relatively well even with a moderate amount of neurons [7, 42, 43]. One can also extend the derivation to other activation functions, which may lead to explicit expressions for the kernel in some cases [19].

**NTK for fully-connected deep ReLU networks.** We define a fully-connected neural network by $f(x; \theta) = \sqrt{\frac{2}{m_n}} \langle w^{n+1}, a^n \rangle$, with $a^1 = \sigma(W^1 x)$, and

$$a^k = \sigma \left( \sqrt{\frac{2}{m_{k-1}}} W^k a^{k-1} \right), \quad k = 2, \ldots, n,$$

where $W^k \in \mathbb{R}^{m_k \times m_{k-1}}$ and $w^{n+1} \in \mathbb{R}^{m_n}$ are initialized with i.i.d. $\mathcal{N}(0, 1)$ entries, and $\sigma(u) = (u)_+$ is the ReLU activation and is applied element-wise. Following [27], the corresponding NTK is defined recursively by $K(x, x') = K_n(x, x')$ with $K_0(x, x') = \Sigma_0(x, x') = x^\top x'$, and for $k \geq 1$,

$$\Sigma_k(x, x') = 2 \, \mathbb{E}_{(u,v) \sim \mathcal{N}(0, B_k)}[\sigma(u)\sigma(v)]$$
$$K_k(x, x') = \Sigma_k(x, x') + 2 K_{k-1}(x, x') \, \mathbb{E}_{(u,v) \sim \mathcal{N}(0, B_k)}[\sigma'(u)\sigma'(v)],$$

where $B_k = \begin{pmatrix} \Sigma_{k-1}(x,x) & \Sigma_{k-1}(x,x') \\ \Sigma_{k-1}(x,x') & \Sigma_{k-1}(x',x') \end{pmatrix}$. Using a change of variables and definitions of arc-cosine kernels of degrees 0 and 1 [17], it is easy to show that

$$2\,\mathbb{E}_{(u,v)\sim\mathcal{N}(0,B_k)}[\sigma(u)\sigma(v)] = \sqrt{\Sigma_{k-1}(x,x)\Sigma_{k-1}(x',x')}\,\kappa_1\left(\frac{\Sigma_{k-1}(x,x')}{\sqrt{\Sigma_{k-1}(x,x)\Sigma_{k-1}(x',x')}}\right) \quad (6)$$

$$2\,\mathbb{E}_{(u,v)\sim\mathcal{N}(0,B_k)}[\sigma'(u)\sigma'(v)] = \kappa_0\left(\frac{\Sigma_{k-1}(x,x')}{\sqrt{\Sigma_{k-1}(x,x)\Sigma_{k-1}(x',x')}}\right), \quad (7)$$

where $\kappa_0$ and $\kappa_1$ are defined in (5).

**Feature maps construction.**  We now provide a reformulation of the previous kernel in terms of explicit feature maps, which provides a representation of the data and makes our study of stability in Section 4 more convenient. For a given input Hilbert space $\mathcal{H}$, we denote by $\varphi_{\mathcal{H},1} : \mathcal{H} \to \mathcal{H}_1$ the kernel mapping into the RKHS $\mathcal{H}_1$ for the kernel $(z,z') \in \mathcal{H}^2 \mapsto \|z\|\|z'\|\kappa_1(\langle z,z'\rangle/\|z\|\|z'\|)$, and by $\varphi_{\mathcal{H},0} : \mathcal{H} \to \mathcal{H}_0$ the kernel mapping into the RKHS $\mathcal{H}_0$ for the kernel $(z,z') \in \mathcal{H}^2 \mapsto \kappa_0(\langle z,z'\rangle/\|z\|\|z'\|)$. We will abuse notation and hide the input space, simply writing $\varphi_1$ and $\varphi_0$.

**Lemma 1** (NTK feature map for fully-connected network). *The NTK for the fully-connected network can be defined as $K(x,x') = \langle\Phi_n(x),\Phi_n(x')\rangle$, with $\Phi_0(x) = \Psi_0(x) = x$ and for $k \geq 1$,*

$$\Psi_k(x) = \varphi_1(\Psi_{k-1}(x))$$
$$\Phi_k(x) = \begin{pmatrix} \varphi_0(\Psi_{k-1}(x)) \otimes \Phi_{k-1}(x) \\ \varphi_1(\Psi_{k-1}(x)) \end{pmatrix},$$

*where $\otimes$ is the tensor product.*

## 2.2  Neural tangent kernel for convolutional networks

In this section we study NTKs for convolutional networks (CNNs) on signals, focusing on the ReLU activation. We consider signals in $\ell^2(\mathbb{Z}^d, \mathbb{R}^{m_0})$, that is, signals $x[u]$ with $u \in \mathbb{Z}^d$ denoting the location, $x[u] \in \mathbb{R}^{m_0}$, and $\sum_{u\in\mathbb{Z}^d}\|x[u]\|^2 < \infty$ (for instance, $d = 2$ and $m_0 = 3$ for RGB images). The infinite support allows us to avoid dealing with boundary conditions when considering deformations and pooling. The precise study of $\ell^2$ membership is deferred to Section 4.

**Patch extraction and pooling operators $P^k$ and $A^k$.**  Following [11], we define two linear operators $P^k$ and $A^k$ on $\ell^2(\mathbb{Z}^d)$ for extracting patches and performing (linear) pooling at layer $k$, respectively. For an $\mathcal{H}$-valued signal $x[u]$, $P^k$ is defined by $P^k x[u] = |S_k|^{-1/2}(x[u+v])_{v\in S_k} \in \mathcal{H}^{|S_k|}$, where $S_k$ is a finite subset of $\mathbb{Z}^d$ defining the patch shape (*e.g.*, a 3x3 box). Pooling is defined as a convolution with a linear filter $h_k[u]$, *e.g.*, a Gaussian filter at scale $\sigma_k$ as in [11], that is, $A^k x[u] = \sum_{v\in\mathbb{Z}^d} h_k[u-v]x[v]$. In this discrete setting, we can easily include a downsampling operation with factor $s_k$ by changing the definition of $A^k$ to $A^k x[u] = \sum_{v\in\mathbb{Z}^d} h_k[s_k u - v]x[v]$ (in particular, if $h_k$ is a Dirac at 0, we obtain a CNN with "strided convolutions"). In fact, our NTK derivation supports general linear operators $A^k : \ell^2(\mathbb{Z}^d) \to \ell^2(\mathbb{Z}^d)$ on scalar signals.

For defining the NTK feature map, we also introduce the following non-linear point-wise operator $M$, given for two signals $x,y$, by

$$M(x,y)[u] = \begin{pmatrix} \varphi_0(x[u]) \otimes y[u] \\ \varphi_1(x[u]) \end{pmatrix}, \quad (8)$$

where $\varphi_{0/1}$ are kernel mappings of arc-cosine 0/1 kernels, as defined in Section 2.1.

**CNN definition and NTK.**  We consider a network $f(x;\theta) = \sqrt{\frac{2}{m_n}}\langle w^{n+1}, a^n\rangle_{\ell^2}$, with

$$\tilde{a}^k[u] = \begin{cases} W^1 P^1 x[u], & \text{if } k = 1, \\ \sqrt{\frac{2}{m_{k-1}}} W^k P^k a^{k-1}[u], & \text{if } k \in \{2,\dots,n\}, \end{cases}$$
$$a^k[u] = A^k \sigma(\tilde{a}^k)[u], \quad k = 1,\dots,n,$$

where $W^k \in \mathbb{R}^{m_k \times m_{k-1}|S_k|}$ and $w^n \in \ell^2(\mathbb{Z}^d, \mathbb{R}^{m_n})$ are initialized with $\mathcal{N}(0,1)$ entries, and $\sigma(\tilde{x}^k)$ denotes the signal with $\sigma$ applied element-wise to $\tilde{x}^k$. We are now ready to state our result on the NTK for this model.

**Proposition 2** (NTK feature map for CNN). *The NTK for the above CNN, obtained when the number of feature maps $m_1, \ldots, m_n \to \infty$ (sequentially), is given by $K(x, x') = \langle \Phi(x), \Phi(x') \rangle_{\ell^2(\mathbb{Z}^d)}$, with $\Phi(x)[u] = A^n M(x_n, y_n)[u]$, where $y_n$ and $x_n$ are defined recursively for a given input $x$ by $y_1[u] = x_1[u] = P^1 x[u]$, and for $k \geq 2$,*

$$x_k[u] = P^k A^{k-1} \varphi_1(x_{k-1})[u]$$
$$y_k[u] = P^k A^{k-1} M(x_{k-1}, y_{k-1})[u],$$

*with the abuse of notation $\varphi_1(x)[u] = \varphi_1(x[u])$ for a signal $x$.*

The proof is given in Appendix A.2, where we also show that in the over-parameterization limit, the pre-activations $\tilde{a}_i^k[u]$ tend to a Gaussian process with covariance $\Sigma^k(x, u; x', u') = \langle x_k[u], x'_k[u'] \rangle$ (this is related to recent papers [24, 40] studying Gaussian process limits of Bayesian convolutional networks). The proof is by induction and relies on similar arguments to [27] for fully-connected networks, in addition to exploiting linearity of the operators $P^k$ and $A^k$, as well as recursive feature maps for hierarchical kernels. The recent papers [3, 51] also study NTKs for certain convolutional networks; in contrast to these works, our derivation considers general signals in $\ell^2(\mathbb{Z}^d)$, supports intermediate pooling or downsampling by changing $A^k$, and provides a more intuitive construction through kernel mappings and the operators $P^k$ and $A^k$. Note that the feature maps $x_k$ are defined independently from the $y_k$, and in fact correspond to more standard multi-layer deep kernel machines [11, 17, 19, 33] or covariance functions of certain deep Bayesian networks [24, 28, 36, 40]. They can also be seen as the feature maps of the limiting kernel that arises when only training weights in the last layer and fixing other layers at initialization (see, *e.g.*, [19]).

# 3 Two-Layer Networks

In this section, we study smoothness and approximation properties of the RKHS defined by neural tangent kernels for two-layer networks. For ReLU activations, we show that the NTK kernel mapping is not Lipschitz, but satisfies a weaker smoothness property. In Section 3.2, we characterize the RKHS for ReLU activations and study its approximation properties and benefits. Finally, we comment on the use of other activations in Section 3.3.

## 3.1 Smoothness of two-layer ReLU networks

Here we study the RKHS $\mathcal{H}$ of the NTK for two-layer ReLU networks, defined in (3), focusing on smoothness properties of the kernel mapping, denoted $\Phi(\cdot)$. Recall that smoothness of the kernel mapping guarantees smoothness of functions $f \in \mathcal{H}$, through the relation

$$|f(x) - f(y)| \leq \|f\|_{\mathcal{H}} \|\Phi(x) - \Phi(y)\|_{\mathcal{H}}. \tag{9}$$

We begin by showing that the kernel mapping for the NTK is not Lipschitz. This is in contrast to the kernel $\kappa_1$ in (5), obtained by fixing the weights in the first layer and training only the second layer weights ($\kappa_1$ is 1-Lipschitz by [11, Lemma 1]).

**Proposition 3** (Non-Lipschitzness). *The kernel mapping $\Phi(\cdot)$ of the two-layer NTK is not Lipschitz:*

$$\sup_{x,y} \frac{\|\Phi(x) - \Phi(y)\|_{\mathcal{H}}}{\|x - y\|} \to +\infty.$$

*This is true even when looking only at points $x, y$ on the sphere. It follows that the RKHS $\mathcal{H}$ contains unit-norm functions with arbitrarily large Lipschitz constant.*

Note that the instability is due to $\varphi_0$, which comes from gradients w.r.t. first layer weigts. We now show that a weaker guarantee holds nevertheless, resembling 1/2-Hölder smoothness.

**Proposition 4** (Smoothness for ReLU NTK). *We have the following smoothness properties:*

*1. For $x, y$ such that $\|x\| = \|y\| = 1$, the kernel mapping $\varphi_0$ satisfies $\|\varphi_0(x) - \varphi_0(y)\| \leq \sqrt{\|x - y\|}$.*

2. *For general non-zero $x, y$, we have $\|\varphi_0(x) - \varphi_0(y)\| \leq \sqrt{\frac{1}{\min(\|x\|, \|y\|)}} \|x - y\|$.*

3. *The kernel mapping $\Phi$ of the NTK then satisfies*

$$\|\Phi(x) - \Phi(y)\| \leq \sqrt{\min(\|x\|, \|y\|)\|x - y\|} + 2\|x - y\|.$$

We note that while such smoothness properties apply to the functions in the RKHS of the studied limiting kernels, the neural network functions obtained at finite width and their linearizations around initialization are not in the RKHS and thus may not preserve such smoothness properties, despite preserving good generalization properties, as in random feature models [7, 43]. This discrepancy may be a source of instability to adversarial perturbations.

## 3.2 Approximation properties for the two-layer ReLU NTK

In the previous section, we found that the NTK $\kappa$ for two-layer ReLU networks yields weaker smoothness guarantees compared to the kernel $\kappa_1$ obtained when the first layer is fixed. We now show that the NTK has better approximation properties, by studying the RKHS through a spectral decomposition of the kernel and the decay of the corresponding eigenvalues. This highlights a tradeoff between smoothness and approximation.

The next proposition gives the Mercer decomposition of the NTK $\kappa(\langle x, u \rangle)$ in (4), where $x, y$ are in the $p-1$ sphere $\mathbb{S}^{p-1} = \{x \in \mathbb{R}^p : \|x\| = 1\}$. The decomposition is given in the basis of spherical harmonics, as is common for dot-product kernels [46, 47], and our derivation uses results by Bach [6] on similar decompositions of positively homogeneous activations of the form $\sigma_\alpha(u) = (u)_+^\alpha$. See Appendix C for background and proofs.

**Proposition 5** (Mercer decomposition of ReLU NTK). *For any $x, y \in \mathbb{S}^{p-1}$, we have the following decomposition of the NTK $\kappa$:*

$$\kappa(\langle x, y \rangle) = \sum_{k=0}^{\infty} \mu_k \sum_{j=1}^{N(p,k)} Y_{k,j}(x) Y_{k,j}(y), \tag{10}$$

*where $Y_{k,j}, j = 1, \ldots, N(p, k)$ are spherical harmonic polynomials of degree $k$, and the non-negative eigenvalues $\mu_k$ satisfy $\mu_0, \mu_1 > 0$, $\mu_k = 0$ if $k = 2j + 1$ with $j \geq 1$, and otherwise $\mu_k \sim C(p)k^{-p}$ as $k \to \infty$, with $C(p)$ a constant depending only on $p$. Then, the RKHS is described by:*

$$\mathcal{H} = \left\{ f = \sum_{k \geq 0, \mu_k \neq 0} \sum_{j=1}^{N(p,k)} a_{k,j} Y_{k,j}(\cdot) \quad s.t. \quad \|f\|_{\mathcal{H}}^2 := \sum_{k \geq 0, \mu_k \neq 0} \sum_{j=1}^{N(p,k)} \frac{a_{k,j}^2}{\mu_k} < \infty \right\}. \tag{11}$$

The zero eigenvalues prevent certain functions from belonging to the RKHS, namely those with non-zero Fourier coefficients on the corresponding basis elements (note that adding a bias may prevent such zero eigenvalues [9]). Here, a sufficient condition for all such coefficients to be zero is that the function is even [6]. Note that for the arc-cosine 1 kernel $\kappa_1$, we have a faster decay $\mu_k = O(k^{-p-2})$, leading to a "smaller" RKHS (see Lemma 17 in Appendix C and [6]). Moreover, the $k^{-p}$ asymptotic equivalent comes from the term $u\kappa_0(u)$ in the definition (4) of $\kappa$, which comes from gradients of first layer weights; the second layer gradients yield $\kappa_1$, whose contribution to $\mu_k$ becomes negligible for large $k$. We use an identity also used in the recent paper [25] which compares similar kernels in a specific high-dimensional limit for generic activations; in contrast to [25], we focus on ReLUs and study eigenvalue decays in finite dimension. We note that our decomposition uses a uniform distribution on the sphere, which allows a precise study of eigenvalues and approximation properties of the RKHS using spherical harmonics. When the data distribution is also uniform on the sphere, or absolutely continuous w.r.t. the uniform distribution, our obtained eigenvalues are closely related to those of integral operators for learning problems, which can determine, *e.g.*, non-parametric rates of convergence (*e.g.*, [14, 23]) as well as degrees-of-freedom quantities for kernel approximation (*e.g.*, [7, 43]). Such quantities often depend on the eigenvalue decay of the integral operator, which can be obtained from $\mu_k$ after taking multiplicity into account. This is also related to the rate of convergence of gradient descent in the lazy training regime, which depends on the minimum eigenvalue of the empirical kernel matrix in [16, 20, 21].

We now provide sufficient conditions for a function $f : \mathbb{S}^{p-1} \to \mathbb{R}$ to be in $\mathcal{H}$, as well as rates of approximation of Lipschitz functions on the sphere, adapting results of [6] (specifically Proposition 2 and 3 in [6]) to our NTK setting.

**Corollary 6** (Sufficient condition for $f \in \mathcal{H}$). *Let $f : \mathbb{S}^{p-1} \to \mathbb{R}$ be an even function such that all i-th order derivatives exist and are bounded by $\eta$ for $0 \leq i \leq s$, with $s \geq p/2$. Then $f \in \mathcal{H}$ with $\|f\|_{\mathcal{H}} \leq C(p)\eta$, where $C(p)$ is a constant that only depends on $p$.*

**Corollary 7** (Approximation of Lipschitz functions). *Let $f : \mathbb{S}^{p-1} \to \mathbb{R}$ be an even function such that $f(x) \leq \eta$ and $|f(x) - f(y)| \leq \eta\|x - y\|$, for all $x, y \in \mathbb{S}^{p-1}$. There is a function $g \in \mathcal{H}$ with $\|g\|_{\mathcal{H}} \leq \delta$, where $\delta$ is larger than a constant depending only on $p$, such that*

$$\sup_{x \in \mathbb{S}^{p-1}} |f(x) - g(x)| \leq C(p)\eta \left(\frac{\delta}{\eta}\right)^{-1/(p/2-1)} \log\left(\frac{\delta}{\eta}\right).$$

For both results, there is an improvement over $\kappa_1$, for which Corollary 6 requires $s \geq p/2 + 1$ bounded derivatives, and Corollary 7 leads to a weaker rate in $(\delta/\eta)^{-1/(p/2)}$ (see [6, Propositions 2 and 3] with $\alpha = 1$). These results show that in the over-parameterized regime of the NTK, training multiple layers leads to better approximation properties compared to only training the last layer, which corresponds to using $\kappa_1$ instead of $\kappa$. In the different regime of "convex neural networks" (*e.g.*, [6, 45]) where neurons can be selected with a sparsity-promoting penalty, the approximation rates shown in [6] for ReLU networks are also weaker than for the NTK in the worst case (though the regime presents benefits in terms of adaptivity), suggesting that perhaps in some situations the "lazy" regime of the NTK could be preferred over the regime where neurons are selected using sparsity.

**Homogeneous case.** When inputs do not lie on the sphere $\mathbb{S}^{p-1}$ but in $\mathbb{R}^p$, the NTK for two-layer ReLU networks takes the form of a homogeneous dot-product kernel (3), which defines a different RKHS $\bar{\mathcal{H}}$ that we characterize below in terms of the RKHS $\mathcal{H}$ of the NTK on the sphere.

**Proposition 8** (RKHS of the homogeneous NTK). *The RKHS $\bar{\mathcal{H}}$ of the kernel $K(x, x') = \|x\|\|x'\|\kappa(\langle x, x'\rangle/\|x\|\|x'\|)$ on $\mathbb{R}^p$ consists of functions of the form $f(x) = \|x\|g(x/\|x\|)$ with $g \in \mathcal{H}$, where $\mathcal{H}$ is the RKHS on the sphere, and we have $\|f\|_{\bar{\mathcal{H}}} = \|g\|_{\mathcal{H}}$.*

Note that while such a restriction to homogeneous functions may be limiting, one may easily obtain non-homogeneous functions by considering an augmented variable $z = (x^\top, R)^\top$ and defining $f(x) = \|z\|g(z/\|z\|)$, where $g$ is now defined on the $p$-sphere $\mathbb{S}^p$. When inputs are in a ball of radius $R$, this reformulation preserves regularity properties (see [6, Section 3]).

### 3.3 Smoothness with other activations

In this section, we look at smoothness of two-layer networks with different activation functions. Following the derivation for the ReLU in Section 2.1, the NTK for a general activation $\sigma$ is given by

$$K_\sigma(x, x') = \langle x, x'\rangle \, \mathbb{E}_{w \sim \mathcal{N}(0,1)}[\sigma'(\langle w, x\rangle)\sigma'(\langle w, x'\rangle)] + \mathbb{E}_{w \sim \mathcal{N}(0,1)}[\sigma(\langle w, x\rangle)\sigma(\langle w, x'\rangle)].$$

We then have the following the following result.

**Proposition 9** (Lipschitzness for smooth activations). *Assume that $\sigma$ is twice differentiable and that the quantities $\gamma_j := \mathbb{E}_{u \sim \mathcal{N}(0,1)}[(\sigma^{(j)}(u))^2]$ for $j = 0, 1, 2$ are bounded, with $\gamma_0 > 0$. Then, for $x, y$ on the unit sphere, the kernel mapping $\Phi_\sigma$ of $K_\sigma$ satisfies*

$$\|\Phi_\sigma(x) - \Phi_\sigma(y)\| \leq \sqrt{(\gamma_0 + \gamma_1) \max\left(1, \frac{2\gamma_1 + \gamma_2}{\gamma_0 + \gamma_1}\right)} \cdot \|x - y\|.$$

The proof uses results from [19] on relationships between activations and the corresponding kernels, as well as smoothness results for dot-product kernels in [11] (see Appendix B.3). If, for instance, we consider the exponential activation $\sigma(u) = e^{u-2}$, we have $\gamma_j = 1$ for all $j$ (using results from [19]), so that the kernel mapping is Lipschitz with constant $\sqrt{3}$. For the soft-plus activation $\sigma(u) = \log(1 + e^u)$, we may evaluate the integrals numerically, obtaining $(\gamma_0, \gamma_1, \gamma_2) \approx (2.31, 0.74, 0.11)$, so that the kernel mapping is Lipschitz with constant $\approx 1.75$.

## 4 Deep Convolutional Networks

In this section, we study smoothness and stability properties of the NTK kernel mapping for convolutional networks with ReLU activations. In order to properly define deformations, we consider

continuous signals $x(u)$ in $L^2(\mathbb{R}^d)$ instead of $\ell^2(\mathbb{Z}^d)$ (*i.e.*, we have $\|x\|^2 := \int \|x(u)\|^2 du < \infty$), following [11, 35]. The goal of deformation stability guarantees is to ensure that the data representation (in this case, the kernel mapping $\Phi$) does not change too much when the input signal is slightly deformed, for instance with a small translation or rotation of an image—a useful inductive bias for natural signals. For a $C^1$-diffeomorphism $\tau : \mathbb{R}^d \to \mathbb{R}^d$, denoting $L_\tau x(u) = x(u - \tau(u))$ the action operator of the diffeomorphism, we will show a guarantee of the form

$$\|\Phi(L_\tau x) - \Phi(x)\| \leq (\omega(\|\nabla\tau\|_\infty) + C\|\tau\|_\infty)\|x\|,$$

where $\|\nabla\tau\|_\infty$ is the maximum operator norm of the Jacobian $\nabla\tau(u)$ over $\mathbb{R}^d$, $\|\tau\|_\infty = \sup_u |\tau(u)|$, $\omega$ is an increasing function and $C$ a positive constant. The second term controls translation invariance, and $C$ typically decreases with the scale of the last pooling layer ($\sigma_n$ below), while the first term controls deformation stability, since $\|\nabla\tau\|_\infty$ measures the "size" of deformations. The function $\omega(t)$ is typically a linear function of $t$ in other settings [11, 35], here we will obtain a faster growth of order $\sqrt{t}$ for small $t$, due to the weaker smoothness that arises from the arc-cosine 0 kernel mappings.

**Properties of the operators.** In this continuous setup, $P^k$ is now given for a signal $x \in L^2$ by $P^k x(u) = \lambda(S_k)^{-1/2}(x(u + v))_{v \in S_k}$, where $\lambda$ is the Lebesgue measure. We then have $\|P^k x\| = \|x\|$, and considering normalized Gaussian pooling filters, we have $\|A^k x\| \leq \|x\|$ by Young's inequality [11]. The non-linear operator $M$ is defined point-wise analogously to (8), and satisfies $\|M(x,y)\|^2 = \|x\|^2 + \|y\|^2$. We thus have that the feature maps in the continuous analog of the NTK construction in Proposition 2 are in $L^2$ as long as $x$ is in $L^2$. Note that this does not hold for some smooth activations, where $\|M(x,y)(u)\|$ may be a positive constant even when $x(u) = y(u) = 0$, leading to unbounded $L^2$ norm for $M(x,y)$. The next lemma studies the smoothness of $M$, extending results from Section 3.1 to signals in $L^2$.

**Lemma 10** (Smoothness of operator $M$). *For two signals $x, y \in L^2(\mathbb{R}^d)$, we have*

$$\|M(x,y) - M(x',y')\| \leq \sqrt{\min(\|y\|, \|y'\|)\|x - x'\|} + \|x - x'\| + \|y - y'\|. \qquad (12)$$

**Assumptions on architecture.** Following [11], we introduce an initial pooling layer $A^0$, corresponding to an anti-aliasing filter, which is necessary for stability and is a reasonable assumption given that in practice input signals are discrete, with high frequencies typically filtered by an acquisition device. Thus, we consider the kernel representation $\Phi_n(x) := \Phi(A^0 x)$, with $\Phi$ as in Proposition 2. We also assume that patch sizes are controlled by the scale of pooling filters, that is

$$\sup_{v \in S_k} |v| \leq \beta \sigma_{k-1}, \qquad (13)$$

for some constant $\beta$, where $\sigma_{k-1}$ is the scale of the pooling operation $A^{k-1}$, which typically increases exponentially with depth, corresponding to a fixed downsampling factor at each layer in the discrete case. By a simple induction, we can show the following.

**Lemma 11** (Norm and smoothness of $\Phi_n$). *We have $\|\Phi_n(x)\| \leq \sqrt{n+1}\|x\|$, and*

$$\|\Phi_n(x) - \Phi_n(x')\| \leq (n+1)\|x - x'\| + O(n^{5/4})\sqrt{\|x\|\|x - x'\|}.$$

**Deformation stability bound.** We now present our main guarantee on deformation stability for the NTK kernel mapping (the proof is given in Appendix B).

**Proposition 12** (Stability of NTK). *Let $\Phi_n(x) = \Phi(A^0 x)$, and assume $\|\nabla\tau\|_\infty \leq 1/2$. We have the following stability bound:*

$$\|\Phi_n(L_\tau x) - \Phi_n(x)\| \leq \left( C(\beta)^{1/2} C n^{7/4} \|\nabla\tau\|_\infty^{1/2} + C(\beta)C'n^2\|\nabla\tau\|_\infty + \sqrt{n+1}\frac{C''}{\sigma_n}\|\tau\|_\infty \right)\|x\|,$$

*where $C, C', C''$ are constants depending only on $d$, and $C(\beta)$ also depends on $\beta$ defined in* (13).

Compared to the bound in [11], the first term shows weaker stability due to faster growth with $\|\nabla\tau\|_\infty$, which comes from (12). The dependence on the depth $n$ is also poorer ($n^2$ instead of $n$), however note that in contrast to [11], the norm and smoothness constants of $\Phi_n(x)$ in Lemma 11 grow with $n$ here, partially explaining this gap. We also note that choosing small $\beta$ (*i.e.*, small patches in a discrete setting) is more helpful to improve stability than a small number of layers $n$, given that $C(\beta)$ increases polynomially with $\beta$, while $n$ typically decreases logarithmically with $\beta$ when one seeks a fixed target level of translation invariance (see [11, Section 3.2]).

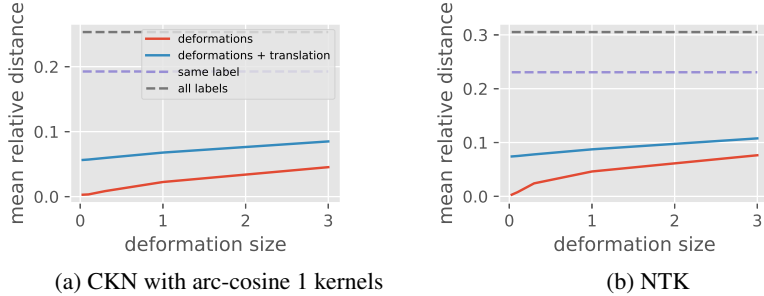

| | |
|---|---|
| (a) CKN with arc-cosine 1 kernels | (b) NTK |

Figure 1: Geometry of kernel mapping for CKN and NTK convolutional kernels, on digit images and their deformations from the Infinite MNIST dataset [32]. The curves show average relative distances of a single digit to its deformations, combinations of translations and deformations, digits of the same label, and digits of any label. See Appendix D for more details on the experimental setup.

By fixing weights of all layers but the last, we would instead obtain feature maps of the form $A^n x_n$ (using notation from Proposition 2), which satisfy the improved stability guarantee of [11]. The question of approximation for the deep convolutional case is more involved and left for future work, but it is reasonable to expect that the RKHS for the NTK is at least as large as that of the simpler kernel with fixed layers before the last, given that the latter appears as one of the terms in the NTK. This again hints at a tradeoff between stability and approximation, suggesting that one may be able to learn less stable but more discriminative functions in the NTK regime by training all layers.

**Numerical experiments.** We now study numerically the stability of (exact) kernel mapping representations for convolutional networks with 2 hidden convolutional layers. We consider both a convolutional kernel network (CKN, [11]) with arc-cosine kernels of degree 1 on patches (corresponding to the kernel obtained when only training the last layer and keeping previous layers fixed) and the corresponding NTK. Figure 1 shows the resulting average distances, when considering collections of digits and deformations thereof. In particular, we find that for small deformations, the distance to the original image tends to grow more quickly for the NTK compared to the CKN, as the theory suggests (a square-root growth rate rather than a linear one). Note also that the relative distances are generally larger for the NTK than for the CKN, suggesting the CKN may be more smooth.

## 5   Discussion

In this paper, we have studied the inductive bias of the "lazy training" regime for over-parameterized neural networks, by considering the neural tangent kernel of different architectures, and analyzing properties of the corresponding RKHS, which characterizes the functions that can be learned efficiently in this regime. We find that the NTK for ReLU networks has better approximation properties compared to other neural network kernels, but weaker smoothness properties, although these can still guarantee a form of stability to deformations for CNN architectures, providing an important inductive bias for natural signals. While these properties may help obtain better performance when large amounts of data are available, they can also lead to a poorer estimation error when data is scarce, a setting in which smoother kernels or better regularization strategies may be helpful.

It should be noted that while our study of functions in the RKHS may determine what target functions can be learned by over-parameterized networks, the obtained networks with finite neurons do not belong to the same RKHS, and hence may be less stable than such target functions, at least outside of the training data, due to approximations both in the linearization (1) and between the finite neuron and limiting kernels. Additionally, approximation of certain non-smooth functions in this regime may require a very large number of neurons [52]. Finally, we note that while this "lazy" regime is interesting and could partly explain the success of deep learning methods, it does not explain, for instance, the common behavior in early layers where neurons move to select useful features in the data, such as Gabor filters, as pointed out in [16]. In particular, such a behavior might provide better statistical efficiency by adapting to simple structures in the data (see, *e.g.*, [6]), something which is not captured in a kernel regime like the NTK. It would be interesting to study inductive biases in a regime somewhere in between, where neurons may move at least in the first few layers.

**Acknowledgments**

This work was supported by the ERC grant number 714381 (SOLARIS project), the ANR 3IA MIAI@Grenoble Alpes, and by the MSR-Inria joint centre. The authors thank Francis Bach and Lénaïc Chizat for useful discussions.

## Footnotes

[2]While we use gradients in our notations, we note that weak differentiability (*e.g.*, with ReLU activations) is sufficient when studying the limiting NTK [27].

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
