[Supplementary Material]

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

# A  Proofs of NTK derivations

## A.1  Proof of Lemma 1

*Proof of Lemma 1.* By induction, using (6) and (7) and the corresponding definitions of $\varphi_1, \varphi_0$, we can write

$$2\,\mathbb{E}_{(u,v)\sim\mathcal{N}(0,B_k)}[\sigma(u)\sigma(v)] = \langle \varphi_1(\Psi_{k-1}(x)), \varphi_1(\Psi_{k-1}(x')) \rangle$$
$$2\,\mathbb{E}_{(u,v)\sim\mathcal{N}(0,B_k)}[\sigma'(u)\sigma'(v)] = \langle \varphi_0(\Psi_{k-1}(x)), \varphi_0(\Psi_{k-1}(x')) \rangle.$$

The result follows by using the following relation, given three pairs of vectors $(x, x')$, $(y, y')$ and $(z, z')$ in arbitrary Hilbert spaces:

$$\langle x, x' \rangle + \langle y, y' \rangle \langle z, z' \rangle = \langle \begin{pmatrix} y \otimes z \\ x \end{pmatrix}, \begin{pmatrix} y' \otimes z' \\ x' \end{pmatrix} \rangle$$

$\square$

## A.2  Proof of Proposition 2 (NTK for CNNs)

In this section, we will denote by $x_k, y_k$ (resp $x'_k, y'_k$) the feature maps associated to an input $x$ (resp $x'$), as defined in Proposition 2. We follow the proofs of Jacot et al. [27, Proposition 1 and Theorem 1].

We begin by proving the following lemma, which characterizes the Gaussian process behavior of the pre-activations $\tilde{a}_i^k[u]$, seen as a function of $x$ and $u$, in the over-parameterization limit.

**Lemma 13.** *As $m_1, \ldots, m_{n-1} \to \infty$, the pre-activations $\tilde{a}_i^k[u]$ for $k = 1, \ldots, n$ tend (in law) to i.i.d. centered Gaussian processes with covariance*

$$\Sigma^k(x, u; x', u') = \langle x_k[u], x'_k[u'] \rangle. \tag{14}$$

*Proof.* We show this by induction. For $k = 1$, $\tilde{a}_i^1[u]$ is clearly Gaussian, and we have

$$\Sigma^1(x, u; x', u') = \mathbb{E}[\tilde{a}_i^1[u]\tilde{a}_i'^1[u']]$$
$$= \mathbb{E}[(W^1 P^1 x[u])_i (W^1 P^1 x'[u'])_i].$$

Writing $W_{ij}^k \in \mathbb{R}^{|S_k|}$ the vector of weights for the filter associated to the input feature map $j$ and output feature map $i$, we have $(W^1 P^1 x[u])_i = \sum_{j=1}^{m_1} W_{ij}^{1\top} P^1 x_j[u]$. Then we have

$$\Sigma^k(x, u; x', u') = \sum_{j,j'} \mathbb{E}[W_{ij}^{1\top} P^1 x_j[u] P^1 x'_{j'}[u']^\top W_{ij'}^1]$$
$$= \sum_{j,j'} \mathrm{Tr}(\mathbb{E}[W_{ij'}^1, W_{ij}^{1\top}] P^1 x_j[u] P^1 x'_{j'}[u']^\top)$$
$$= \sum_{j} \mathrm{Tr}(P^1 x_j[u] P^1 x'_j[u']^\top) = \langle P^1 x[u], P^1 x'[u'] \rangle = \langle P^1 x_0[u], P^1 x'_0[u'] \rangle,$$

by noticing that $\mathbb{E}[W_{ij'}^1, W_{ij}^{1\top}] = \delta_{j,j'} I_{|S_1|}$.

Now, for $k \geq 2$, we have by similar arguments that conditioned on $a^{k-1}$, $\tilde{a}_i^k[u]$ is Gaussian, with covariance

$$\mathbb{E}[\tilde{a}_i^k[u]\tilde{a}_i'^k[u']|a^{k-1}, a'^{k-1}] = \frac{2}{m_{k-1}} \sum_j \langle P^k a_j^{k-1}[u], P^k a_j'^{k-1}[u'] \rangle.$$

By the inductive hypothesis, $\tilde{a}_j^{k-1}[u]$ as a function of $x$ and $u$ tend to Gaussian processes in the limit $m_1, \ldots, m_{k-2} \to \infty$. By the law of large numbers, we have, as $m_{k-1} \to \infty$,

$$\mathbb{E}[\tilde{a}_i^k[u]\tilde{a}_i'^k[u']|a^{k-1}, a'^{k-1}]$$
$$\to \Sigma^k(x, u; x', u') := 2\,\mathbb{E}_{f\sim GP(0,\Sigma^{k-1})}[\langle P^k A^{k-1}\sigma(f(x))[u], P^k A^{k-1}\sigma(f(x'))[u'] \rangle].$$

Since this covariance is deterministic, the pre-activations $\tilde{a}_i^k[u]$ are also unconditionally a Gaussian process in the limit, with covariance $\Sigma^k$.

Now it remains to show that

$$2\,\mathbb{E}_{f\sim GP(0,\Sigma^{k-1})}[\langle P^k A^{k-1}\sigma(f(x))[u], P^k A^{k-1}\sigma(f(x'))[u']\rangle]$$
$$= \langle P^k A^{k-1}\varphi_1(x_{k-1})[u], P^k A^{k-1}\varphi_1(x'_{k-1})[u']\rangle.$$

Notice that by linearity of $P^k$ and $A^{k-1}$, it suffices to show

$$2\,\mathbb{E}_{f\sim GP(0,\Sigma^{k-1})}[\sigma(f(x))[v]\sigma(f(x'))[v']] = \langle\varphi_1(x_{k-1})[v], \varphi_1(x'_{k-1})[v']\rangle$$
$$= \|x_{k-1}[v]\|\|x'_{k-1}[v']\|\kappa_1\left(\frac{\langle x_{k-1}[v], x_{k-1}[v']\rangle}{\|x_{k-1}[v]\|\|x'_{k-1}[v']\|}\right),$$

for any $v, v'$ (the last equality follows from the definition of $\varphi_1$). Noting that $(\sigma(f(x))[v], \sigma(f(x'))[v']) = (\sigma(f(x)[v]), \sigma(f(x')[v']))$ when $f \sim GP(0, \Sigma^{k-1})$ has Gaussian distribution with zero mean and covariance $\begin{pmatrix} \Sigma^{k-1}(x,v;x,v) & \Sigma^{k-1}(x,v;x',v') \\ \Sigma^{k-1}(x',v';x',v') & \Sigma^{k-1}(x',v';x',v') \end{pmatrix}$, the results follow from (6) and (14) for $k$–1 by the inductive hypothesis. $\square$

We now state and prove a lemma which covers the recursion in the NTK for convolutional layers (*i.e.*, up to the last fully-connected layer).

**Lemma 14.** *As $m_1, \ldots, m_{n-1} \to \infty$, the gradients of the pre-activations, $\nabla_\theta \tilde{a}_i^k[u]$, for $k = 1, \ldots, n$ satisfy*

$$\langle\nabla_\theta \tilde{a}_i^k[u], \nabla_\theta \tilde{a}_{i'}^{\prime k}[u']\rangle \to \delta_{i,i'}\tilde{\Gamma}_\infty^k(x,u;x',u') = \delta_{i,i'}\langle y_k[u], y'_k[u']\rangle.$$

*Proof.* We prove this by induction. For $k = 1$, denoting by $W_i^1$ the $i$th row of $W^1$, we have

$$\langle\nabla_\theta \tilde{a}_i^1[u], \nabla_\theta \tilde{a}_{i'}^{\prime 1}[u']\rangle = \langle\nabla_{W^1}(W^1 P^1 x[u])_i, \nabla_{W^1}(W^1 P^1 x'[u'])_{i'}\rangle$$
$$= \sum_s \langle\nabla_{W_s^1} W_i^1 P^1 x[u], \nabla_{W_s^1} W_{i'}^1 P^1 x'[u']\rangle$$
$$= \delta_{i,i'}\langle P^1 x[u], P^1 x'[u']\rangle.$$

For $k \geq 2$, assume the result holds up to $k$–1. We have

$$\langle\nabla_\theta \tilde{a}_i^k[u], \nabla_\theta \tilde{a}_{i'}^{\prime k}[u']\rangle = \langle\nabla_{W^k} \tilde{a}_i^k[u], \nabla_{W^k} \tilde{a}_{i'}^{\prime k}[u']\rangle + \langle\nabla_{W^{1:k-1}} \tilde{a}_i^k[u], \nabla_{W^{1:k-1}} \tilde{a}_{i'}^{\prime k}[u']\rangle.$$

For the first term, we have, as in the $k = 1$ case,

$$\langle\nabla_{W^k} \tilde{a}_i^k[u], \nabla_{W^k} \tilde{a}_{i'}^{\prime k}[u']\rangle = \frac{2\delta_{i,i'}}{m_{k-1}}\langle P^k a^{k-1}[u], P^k a^{\prime k-1}[u']\rangle$$
$$= \frac{2\delta_{i,i'}}{m_{k-1}}\sum_j \langle P^k a_j^{k-1}[u], P^k a_j^{\prime k-1}[u']\rangle$$
$$= \frac{2\delta_{i,i'}}{m_{k-1}}\sum_j \langle P^k A^{k-1}\sigma(\tilde{a}_j^{k-1})[u], P^k A^{k-1}\sigma(\tilde{a}_j^{\prime k-1})[u']\rangle.$$

When $m_1, \ldots, m_{k-2} \to \infty$, $\tilde{a}_j^{k-1}[u]$ tends to a Gaussian process with covariance $\Sigma^{k-1}$ by Lemma 13, and when $m_{k-1} \to \infty$, the quantity above converges to its expectation:

$$\langle\nabla_{W^k} \tilde{a}_i^k[u], \nabla_{W^k} \tilde{a}_{i'}^{\prime k}[u']\rangle \to 2\delta_{i,i'}\,\mathbb{E}_{f\sim GP(0,\Sigma^{k-1})}[\langle P^k A^{k-1}\sigma(f(x))[u], P^k A^{k-1}\sigma(f(x))[u']\rangle]$$
$$= \delta_{i,i'}\langle P^k A^{k-1}\varphi_1(x_{k-1})[u], P^k A^{k-1}\varphi_1(x'_{k-1})[u']\rangle,$$

by using similar arguments to the proof of Lemma 13.

For the second term, identifying all parameters $W^{1:k-1}$ with a vector $\hat{\theta} \in \mathbb{R}^q$, we have by linearity and the chain rule:

$$\sqrt{m_{k-1}}\nabla_{\hat{\theta}} \tilde{a}_i^k[u] = \sum_j \nabla_{\hat{\theta}}(W_{ij}^{k\top} P^k A^{k-1}\tilde{a}_j^{k-1}[u])$$
$$= \sum_j P^k A^{k-1}y_j^{k-1}[u]^\top \cdot W_{ij}^k \in \mathbb{R}^q,$$

where $y_j^{k-1}[u] := \sqrt{2}\sigma'(\tilde{a}_j^{k-1}[u])\nabla_{\hat\theta}\tilde{a}_j^{k-1}[u] \in \mathbb{R}^q$. Here we have identified $P^k A^{k-1} y_j^{k-1}[u]^\top$ with a matrix in $\mathbb{R}^{q\times|S_k|}$, where columns are given by $|S_k|^{-1/2}A^{k-1}y_j^{k-1}[u+v] \in \mathbb{R}^q$, indexed by $v \in S_k$. We thus have

$$\langle \nabla_{W^{1:k-1}}\tilde{a}_i^k[u], \nabla_{W^{1:k-1}}\tilde{a}_{i'}'^k[u']\rangle = \frac{1}{m_{k-1}}\sum_{j,j'} W_{i,j}^{k\top}\underbrace{(P^k A^{k-1}y_j^{k-1}[u]\cdot P^k A^{k-1}y_{j'}'^{k-1}[u']^\top)}_{=:\Pi^{j,j'}\in\mathbb{R}^{|S_k|\times|S_k|}}W_{i',j'}^k.$$

For $v,v' \in S_k$, when $m_1,\ldots,m_{k-2}\to\infty$ and using the inductive hypothesis, we have

$$\Pi_{v,v'}^{j,j'} = \frac{1}{|S_k|}A^{k-1}y_j^{k-1}[u+v]^\top A^{k-1}y_{j'}'^{k-1}[u'+v']$$

$$\to \delta_{j,j'}\bar\Pi_{v,v'}^j := \frac{\delta_{j,j'}}{|S_k|}\langle A^{k-1}\gamma_j^{k-1}[u+v], A^{k-1}\gamma_j'^{k-1}[u'+v']\rangle,$$

where $\gamma_j^{k-1}[u] := \sqrt{2}\sigma'(\tilde{a}_j^{k-1}[u])y_{k-1}[u]$. Indeed, by linearity it suffices to check that $y_j^{k-1}[u]^\top y_{j'}'^{k-1}[u'] = 2\delta_{j,j'}\sigma'(\tilde{a}_j^{k-1}[u])\sigma'(\tilde{a}_{j'}'^{k-1}[u'])\langle y_{k-1}[u], y_{k-1}'[u']\rangle$ for any $j,j',u,u'$, which is true by the inductive hypothesis. In this same limit (with $m_{k-1}$ fixed), we then have

$$\langle \nabla_{W^{1:k-1}}\tilde{a}_i^k[u], \nabla_{W^{1:k-1}}\tilde{a}_{i'}'^k[u']\rangle \to \frac{1}{m_{k-1}}\sum_j W_{i,j}^{k\top}\bar\Pi^j W_{i',j}^k.$$

When $m_{k-1}\to\infty$, by the law of large numbers, this quantity converges to its expectation:

$$\langle \nabla_{W^{1:k-1}}\tilde{a}_i^k[u], \nabla_{W^{1:k-1}}\tilde{a}_{i'}'^k[u']\rangle \to \mathrm{Tr}(\mathbb{E}[W_{i',1}^k W_{i,1}^{k\top}]\Pi^\infty) = \delta_{i,i'}\,\mathrm{Tr}(\Pi^\infty),$$

where $\Pi^\infty$ is given by

$$\Pi_{v,v'}^\infty = \frac{1}{|S_k|}\langle A^{k-1}\gamma_\infty^{k-1}[u+v], A^{k-1}\gamma_\infty'^{k-1}[u'+v']\rangle,$$

with $\gamma_\infty^{k-1}[u] = \varphi_0(x_{k-1}[u])\otimes y_{k-1}[u]$. Indeed, using Lemma 13 and linearity of $A^{k-1}$, it is enough to check that

$$2\,\mathbb{E}_{f\sim GP(0,\Sigma^{k-1})}[\sigma'(f(x)[u])\sigma'(f(x')[u'])\langle y_{k-1}[u], y_{k-1}'[u']\rangle] = \langle\gamma_\infty^{k-1}[u], \gamma_\infty'^{k-1}[u']\rangle,$$

which holds by definition of $\varphi_0$ and $\Sigma^{k-1}$.

Finally, notice that

$$\mathrm{Tr}(\Pi^\infty) = \frac{1}{|S_k|}\sum_{v\in S_k}\langle A^{k-1}\gamma_\infty^{k-1}[u+v], A^{k-1}\gamma_\infty'^{k-1}[u'+v]\rangle$$

$$= \langle P^k A^{k-1}\gamma_\infty^{k-1}[u], P^k A^{k-1}\gamma_\infty'^{k-1}[u']\rangle.$$

Thus we have

$$\tilde{\Gamma}_\infty(x,u,x',u') = \langle P^k A^{k-1}\varphi_1(x_{k-1})[u], P^k A^{k-1}\varphi_1(x_{k-1}')[u']\rangle + \langle P^k A^{k-1}\gamma_\infty^{k-1}[u], P^k A^{k-1}\gamma_\infty'^{k-1}[u']\rangle$$

$$= \langle P^k A^{k-1}M(x_{k-1}, y_{k-1})[u], P^k A^{k-1}M(x_{k-1}', y_{k-1}')[u']\rangle$$

$$= \langle y_k[u], y_k'[u']\rangle,$$

which concludes the proof.

$\square$

Armed with the two above lemmas, we can now prove Proposition 2 by studying the gradient of the prediction layer.

*Proof of Proposition 2.* We have

$$\langle \nabla_\theta f(x;\theta), \nabla_\theta f(x';\theta)\rangle = \langle \nabla_{w^{n+1}}f(x;\theta), \nabla_{w^{n+1}}f(x';\theta)\rangle + \langle \nabla_{W^{1:n}}f(x;\theta), \nabla_{W^{1:n}}f(x';\theta)\rangle$$

The first term writes

$$\langle \nabla_{w^{n+1}} f(x; \theta), \nabla_{w^{n+1}} f(x'; \theta) \rangle = \frac{2}{m_n} \sum_j \langle a_j^n, a_j'^n \rangle$$

$$= \frac{2}{m_n} \sum_j \sum_u \langle A^n \sigma(\tilde{a}_j^n)[u], A^n \sigma(\tilde{a}_j'^n)[u] \rangle$$

Using similar arguments as in the above proofs and using Lemma 13, as $m_1, \dots, m_n \to \infty$, we have

$$\langle \nabla_{w^{n+1}} f(x; \theta), \nabla_{w^{n+1}} f(x'; \theta) \rangle \to \sum_u \langle A^n \varphi_1(x_n)[u], A^n \varphi_1(x'_n)[u] \rangle = \langle A^n \varphi_1(x_n), A^n \varphi_1(x'_n) \rangle.$$

For the second term, we have

$$\langle \nabla_{W^{1:n}} f(x; \theta), \nabla_{W^{1:n}} f(x'; \theta) = \frac{2}{m_n} \sum_{u,u'} \sum_{j,j'} w_j^{n+1}[u] w_{j'}^{n+1}[u'] \langle \nabla_{W^{1:n}} a_j^n[u], \nabla_{W^{1:n}} a_{j'}'^n[u'] \rangle.$$

We can use similar arguments to the proof of Lemma 14 to show that when $m_1, \dots, m_n \to \infty$, we have

$$\langle \nabla_{W^{1:n}} f(x; \theta), \nabla_{W^{1:n}} f(x'; \theta) \to \sum_{u,u'} \mathbb{E}[w_1^{n+1}[u] w_1^{n+1}[u']] \langle A^n \gamma_n[u], A^n \gamma_n'[u'] \rangle = \langle A^n \gamma_n, A^n \gamma_n' \rangle,$$

where $\gamma_n[u] := \varphi_0(x_n[u]) \otimes y_n[u]$.

The final result follows by combining both terms. $\qquad\qquad\square$

## B   Proofs for Smoothness and Stability to Deformations

### B.1   Proof of Proposition 3

*Proof.* Using notations from Section 2.1, we can write

$$\kappa(u) = u\kappa_0(u) + \kappa_1(u) = \frac{u}{\pi}(\pi - \arccos(u)) + \frac{1}{\pi}(u(\pi - \arccos(u)) + \sqrt{1 - u^2}).$$

For $\|x\| = \|y\| = 1$, and denoting $u = \langle x, y \rangle$, we have

$$\frac{\|\Phi(x) - \Phi(y)\|^2}{\|x - y\|^2} = \frac{2\kappa(1) - 2\kappa(u)}{2 - 2u}$$

$$= \frac{\kappa_0(1) - u\kappa_0(u)}{1 - u} + \frac{\kappa_1(1) - \kappa_1(u)}{1 - u}$$

$$\sim_{u \to 1^-} u\kappa_0'(u) + \kappa_0(u) + \kappa_1'(u) \xrightarrow{u \to 1^-} +\infty,$$

where the equivalent follows from l'Hôpital's rule, and we have $\kappa_0'(u) = 1/\pi\sqrt{1 - u^2} \to +\infty$, while $\kappa_0(1) = \kappa_1(1) = \kappa_1'(1) = 1$. It follows that the supremum over $x, y$ is unbounded.

For the second part, fix an arbitrary $L > 0$. We can find $x, y$ such that $\|\Phi(x) - \Phi(y)\|_{\mathcal{H}} > L\|x - y\|$. Take

$$f = \frac{\Phi(x) - \Phi(y)}{\|\Phi(x) - \Phi(y)\|_{\mathcal{H}}} \in \mathcal{H}.$$

We have $\|f\|_{\mathcal{H}} = 1$ and $f(x) - f(y) = \|\Phi(x) - \Phi(y)\|_{\mathcal{H}} > L\|x - y\|$, so that the Lipschitz constant of $f$ is larger than $L$. $\qquad\square$

### B.2   Proof of Proposition 4 (smoothness of 2-layer ReLU NTK)

*Proof.* Denoting $u = \langle x, y \rangle$, we have

$$\frac{\|\varphi_0(x) - \varphi_0(y)\|^2}{\|x - y\|} = \frac{2\kappa_0(1) - 2\kappa_0(u)}{\sqrt{2 - 2u}}.$$

As a function of $u \in [-1, 1]$, this quantity decreases from 1 to $1/2\pi$, and is thus upper bounded by 1, proving the first part.

Note that if $u, v$ are on the sphere and $\alpha \geq 1$, then $\|u - \alpha v\| \geq \|u - v\|$. This yields

$$\|x - y\| \geq \min(\|x\|, \|y\|)\|\bar{x} - \bar{y}\|,$$

where $\bar{x}, \bar{y}$ denote the normalized vectors. Then, noting that $\varphi_0$ is 0-homogeneous, we have

$$
\begin{aligned}
\frac{\|\varphi_0(x) - \varphi_0(y)\|^2}{\|x - y\|} &= \frac{\|\varphi_0(\bar{x}) - \varphi_0(\bar{y})\|^2}{\|x - y\|} \\
&\leq \frac{\|\varphi_0(\bar{x}) - \varphi_0(\bar{y})\|^2}{\min(\|x\|, \|y\|)\|\bar{x} - \bar{y}\|} \\
&\leq \frac{1}{\min(\|x\|, \|y\|)},
\end{aligned}
$$

by using the previous result on the sphere, and the result for the second part follows.

For the last part, assume $x$ has smaller norm than $y$. We have

$$
\begin{aligned}
\|\Phi(x) - \Phi(y)\| &= \left\| \begin{pmatrix} \varphi_0(x) \otimes x \\ \varphi_1(x) \end{pmatrix} - \begin{pmatrix} \varphi_0(y) \otimes y \\ \varphi_1(y) \end{pmatrix} \right\| \\
&\leq \left\| \begin{pmatrix} \varphi_0(x) \otimes x \\ \varphi_1(x) \end{pmatrix} - \begin{pmatrix} \varphi_0(y) \otimes x \\ \varphi_1(y) \end{pmatrix} \right\| + \left\| \begin{pmatrix} \varphi_0(y) \otimes x \\ \varphi_1(y) \end{pmatrix} - \begin{pmatrix} \varphi_0(y) \otimes y \\ \varphi_1(y) \end{pmatrix} \right\| \\
&= \sqrt{\|x\|^2 \|\varphi_0(x) - \varphi_0(y)\|^2 + \|\varphi_1(x) - \varphi_1(y)\|^2} + \|\varphi_0(y)\| \|x - y\| \\
&\leq \sqrt{\|x\| \|x - y\| + \|x - y\|^2} + \|x - y\| \leq \sqrt{\|x\| \|x - y\|} + 2\|x - y\|,
\end{aligned}
$$

where in the last line we used $\|\varphi_0(y)\| = 1$, $\|\varphi_0(x) - \varphi_0(y)\|^2 \leq \|x - y\|/\|x\|$, as well as $\|\varphi_1(x) - \varphi_1(y)\| \leq \|x - y\|$, which follows from [11, Lemma 1]. We conclude by symmetry. $\square$

### B.3 Proof of Proposition 9 (smooth activations)

*Proof.* We introduce the following kernels defined on the sphere:

$$\kappa_j(\langle x, x' \rangle) = \mathbb{E}_{w \sim \mathcal{N}(0, I)}[\sigma^{(j)}(\langle w, x \rangle)\sigma^{(j)}(\langle w, x' \rangle)].$$

Note that these are indeed dot-product kernels, defined as polynomial expansions in terms of the squared Hermite expansion coefficients of $\sigma^{(j)}$, as shown by Daniely [19] (called "dual activations"). In fact, [19, Lemma 11] also shows that the mapping from activation to dual activation commutes with differentiation, so that $\kappa_j(u) = \kappa_0^{(j)}(u)$, for $u \in (-1, 1)$. The assumption made in this proposition implies that the $j$-th order derivatives of $\kappa_0$ as $u \to 1$ exist, with $\kappa_0^{(j)}(1) = \kappa_j(1) = \gamma_j < +\infty$.

Then, the NTK on the sphere takes the form $K_\sigma(x, x') = \kappa_\sigma(\langle x, x' \rangle)$, where $\kappa_\sigma(u) = u\kappa_1(u) + \kappa_0(u)$. Then, if we consider the kernel $\hat{\kappa}_\sigma(u) = \frac{\kappa_\sigma(u)}{\kappa_\sigma(1)} = \frac{\kappa_\sigma(u)}{\gamma_0 + \gamma_1}$, we have

$$\hat{\kappa}_\sigma(1) = 1 \quad \text{and} \quad \hat{\kappa}'_\sigma(1) = \frac{\kappa_1(1) + \kappa_1'(1) + \kappa_0'(1)}{\gamma_0 + \gamma_1} = \frac{\gamma_2 + 2\gamma_1}{\gamma_0 + \gamma_1}.$$

Applying Lemma 1 of [11] to this kernel, and re-multiplying by $\gamma_0 + \gamma_1$ yields the final result. $\square$

### B.4 Proof of Lemma 10 (smoothness of operator $M$ in $L^2(\mathbb{R}^d)$)

*Proof.* Using similar arguments as in the proof of Proposition 4, we can show that for any $u \in \mathbb{R}^d$

$$
\begin{aligned}
\|M(x, y)(u) - M(x', y')(u)\| &\leq \sqrt{\min(\|y(u)\|, \|y'(u)\|)\|x(u) - x'(u)\|} \\
&\quad + \|x(u) - x'(u)\| + \|y(u) - y'(u)\|
\end{aligned}
$$

Now assume that $\min(\|y\|, \|y'\|) = \|y\|$. By the triangle inequality in $L^2(\mathbb{R}^d)$, we then have

$$\|M(x, y) - M(x', y')\| \leq \sqrt{\int \min(\|y(u)\|, \|y'(u)\|) \|x(u) - x'(u)\| du} + \|x - x'\| + \|y - y'\|$$

$$\leq \sqrt{\int \|y(u)\| \|x(u) - x'(u)\| du} + \|x - x'\| + \|y - y'\|$$

$$\leq \sqrt{\|y\| \|x - x'\|} + \|x - x'\| + \|y - y'\|,$$

where the last inequality follows from Cauchy-Schwarz. We obtain the final result by symmetry. $\quad\square$

## B.5  Proof of Proposition 12 (stability to deformations)

We first recall the following results from [11].

**Lemma 15** ([11]). *Assume $\|\nabla\tau\|_\infty \leq 1/2$, and $\sup_{v \in S_k} |v| \leq \beta\sigma_{k-1}$ for all $k$. We have*

$$\|[P^k A^{k-1}, L_\tau]\| \leq C(\beta)\|\nabla\tau\|_\infty$$

$$\|L^\tau A^n - A^n\| \leq \frac{C_2}{\sigma_n}\|\tau\|_\infty,$$

*where $C(\beta)$ grows with $\beta$ as $\beta^{d+1}$.*

We are now ready to prove Proposition 12.

*Proof.* In order to compare $\Phi_n(L_\tau x)$ and $\Phi_n(x)$, we introduce intermediate sequences of feature maps, denoted $x_k^{(k_0)}$ and $y_k^{(k_0)}$, where the deformation operator $L_\tau$ acts at layer $k_0$. In particular, we denote by $x_k^{(0)}$, $y_k^{(0)}$ the feature maps obtained for the input $L_\tau x$, and if $k_0 \geq 1$, we define $x_k^{(k_0)} = x_k$, $y_k^{(k_0)} = y_k$ for $k \leq k_0$,

$$x_{k_0+1}^{(k_0)} = P^{k_0+1} A^{k_0} L_\tau \varphi_1(x_{k_0})$$

$$y_{k_0+1}^{(k_0)} = P^{k_0+1} A^{k_0} L_\tau M(x_{k_0}, y_{k_0}),$$

and for $k \geq k_0 + 2$,

$$x_k^{(k_0)} = P^k A^{k-1} L_\tau \varphi_1(x_{k-1}^{k_0})$$

$$y_k^{(k_0)} = P^k A^{k-1} L_\tau M(x_{k-1}^{k_0}, y_{k-1}^{k_0}).$$

Then, we have the following

$$\|\Phi_n(L_\tau x) - \Phi_n(x)\| = \|A^n M(x_n^{(0)}, y_n^{(0)}) - A^n M(x_n, y_n)\|$$

$$\leq \|A^n M(x_n^{(0)}, y_n^{(0)}) - A^n L_\tau M(x_n, y_n)\|$$

$$+ \|A^n L_\tau M(x_n, y_n) - A^n M(x_n, y_n)\|$$

$$\leq \|A^n M(x_n^{(0)}, y_n^{(0)}) - A^n L_\tau M(x_n, y_n)\|$$

$$+ \|A^n L_\tau - A^n\| \|M(x_n, y_n)\|.$$

Using Lemma 15, we have

$$\|A^n L_\tau - A^n\| \leq \|[A^n, L_\tau]\| + \|L_\tau A^n - A^n\|$$

$$\leq C(\beta)\|\nabla\tau\|_\infty + \frac{C_2}{\sigma_n}\|\tau\|_\infty.$$

Separately, we have $\|M(x_n, y_n)\|^2 = \|x_n\|^2 + \|y_n\|^2 \leq (n+1)\|x\|^2$, so that

$$\|\Phi_n(L_\tau x) - \Phi_n(x)\| \leq \|A^n M(x_n^{(0)}, y_n^{(0)}) - A^n L_\tau M(x_n, y_n)\|$$

$$+ \sqrt{n+1}\left(C(\beta)\|\nabla\tau\|_\infty + \frac{C_2}{\sigma_n}\|\tau\|_\infty\right)\|x\|.$$

We now bound the first term above by induction. For $n = 1$, we have

$$\|A^1 M(x_1^{(0)}, y_1^{(0)}) - A^1 L_\tau M(x_1, y_1)\| \leq \|A^1 M(x_1^{(0)}, y_1^{(0)}) - A^1 L_\tau M(x_1, y_1)\|$$
$$\leq \|M(x_1^{(0)}, y_1^{(0)}) - L_\tau M(x_1, y_1)\|$$
$$= \|M(L_\tau x, L_\tau x) - L_\tau M(x, y)\| = 0,$$

by noting that $M$ is a point-wise operator, and thus commutes with $L_\tau$. We now assume $n \geq 2$. We have

$$\|A^n M(x_n^{(0)}, y_n^{(0)}) - A^n L_\tau M(x_n, y_n)\| \leq \|M(x_n^{(0)}, y_n^{(0)}) - L_\tau M(x_n, y_n)\|$$
$$= \|M(x_n^{(0)}, y_n^{(0)}) - M(L_\tau x_n, L_\tau y_n)\|$$
$$\leq \sqrt{\|L_\tau y_n\| \|x_n^{(0)} - L_\tau x_n\|}$$
$$+ \|x_n^{(0)} - L_\tau x_n\| + \|y_n^{(0)} - L_\tau y_n\|,$$

where we used Lemma 10 and the fact that $M$ commutes with $L_\tau$.

Now note that since $\|\nabla \tau\|_\infty \leq 1/2$, for any signal $x$ we have

$$\|L_\tau x\|^2 = \int \|x(u - \tau(u))\|^2 du = \int \|x(u)\|^2 |\det(I - \nabla \tau(u))|^{-1} du$$
$$\leq \frac{1}{(1 - \|\nabla \tau\|_\infty)^d} \|x\|^2 \leq 2^d \|x\|^2. \tag{15}$$

Thus, we have $\|L_\tau y_n\| \leq 2^{d/2} \|y_n\| \leq \sqrt{2^d n} \|x\|$. Separately, using the non-expansivity of $\varphi_1$, we have

$$\|x_n^{(0)} - L_\tau x_n\| \leq \sum_{k=1}^{n-1} \|x_n^{(k-1)} - x_n^{(k)}\| + \|x_n^{(n-1)} - L_\tau x_n\|$$
$$\leq \sum_{k=1}^{n} \|x_k^{(k-1)} - L_\tau x_k\|$$
$$= \|P^1 A^0 L_\tau x - L_\tau P^1 A^0 x\| + \sum_{k=2}^{n} \|P^k A^{k-1} L_\tau \varphi_1(x_{k-1}) - L_\tau P^k A^{k-1} \varphi_1(x_{k-1})\|$$
$$\leq \sum_{k=1}^{n} \|[P^k A^{k-1}, L_\tau]\| \|x\|$$
$$\leq C(\beta) n \|\nabla \tau\|_\infty \|x\|,$$

by Lemma 15. We also have

$$\|y_n^{(0)} - L_\tau y_n\| = \|P^n A^{n-1} M(x_{n-1}^{(0)}, y_{n-1}^{(0)}) - L_\tau P^n A^{n-1} M(x_{n-1}, y_{n-1})\|$$
$$\leq \|P^n A^{n-1} M(x_{n-1}^{(0)}, y_{n-1}^{(0)}) - P^n A^{n-1} L_\tau M(x_{n-1}, y_{n-1})\| + \|[P^n A^{n-1}, L_\tau]\| \|M(x_{n-1}, y_{n-1})\|$$
$$\leq \|A^{n-1} M(x_{n-1}^{(0)}, y_{n-1}^{(0)}) - A^{n-1} L_\tau M(x_{n-1}, y_{n-1})\| + C(\beta) \sqrt{n} \|\nabla \tau\|_\infty \|x\|.$$

We have thus shown:

$$\|A^n M(x_n^{(0)}, y_n^{(0)}) - A^n L_\tau M(x_n, y_n)\| \leq \left( 2^{d/4} C(\beta)^{1/2} n^{3/4} \|\nabla \tau\|_\infty^{1/2} + C(\beta)(n + \sqrt{n}) \|\nabla \tau\|_\infty \right) \|x\|$$
$$+ \|A^{n-1} M(x_{n-1}^{(0)}, y_{n-1}^{(0)}) - A^{n-1} L_\tau M(x_{n-1}, y_{n-1})\|.$$

Unrolling the recurrence relation yields

$$\|A^n M(x_n^{(0)}, y_n^{(0)}) - A^n L_\tau M(x_n, y_n)\| \leq \sum_{k=2}^{n} \left( 2^{d/4} C(\beta)^{1/2} k^{3/4} \|\nabla \tau\|_\infty^{1/2} + C(\beta)(k + \sqrt{k}) \|\nabla \tau\|_\infty \right) \|x\|$$
$$\leq \left( C(\beta)^{1/2} C n^{7/4} \|\nabla \tau\|_\infty^{1/2} + C(\beta) C' n^2 \|\nabla \tau\|_\infty \right) \|x\|,$$

where $C, C'$ are absolute constants depending only on $d$.

The final bound becomes

$$\|\Phi_n(L_\tau x) - \Phi_n(x)\| \leq \left( C(\beta)^{1/2} C n^{7/4} \|\nabla \tau\|_\infty^{1/2} + C(\beta) C' n^2 \|\nabla \tau\|_\infty + \sqrt{n+1} \frac{C_2}{\sigma_n} \|\tau\|_\infty \right) \|x\|,$$

with a different constant $C'$.

$\square$

## C Approximation Properties

### C.1 Background on spherical harmonics

In this section, we provide some background on spherical harmonic analysis needed for our study of the RKHS. See [22, 5] for references, as well as [6, Appendix D]. We consider inputs on the $p-1$ sphere $\mathbb{S}^{p-1} = \{x \in \mathbb{R}^p, \|x\| = 1\}$.

We denote by $Y_{kj}(x)$, $j = 1, \ldots, N(p,k)$, the spherical harmonics of degree $k$ on $\mathbb{S}^{p-1}$, where $N(p,k) = \frac{2k+p-2}{k} \binom{k+p-3}{p-2}$. They form an orthonormal basis of $L^2(\mathbb{S}^{p-1}, d\tau)$, where $\tau$ is the uniform measure on the sphere. The index $k$ plays the role of an integer frequency, as in Fourier series. We have the addition formula

$$\sum_{j=1}^{N(p,k)} Y_{k,j}(x) Y_{k,j}(y) = N(p,k) P_k(\langle x, y \rangle), \tag{16}$$

where $P_k$ is the $k$-th Legendre polynomial in dimension $p$ (also known as Gegenbauer polynomials), given by the Rodrigues formula:

$$P_k(t) = (-1/2)^k \frac{\Gamma(\frac{p-1}{2})}{\Gamma(k + \frac{p-1}{2})} (1 - t^2)^{(3-p)/2} \left( \frac{d}{dt} \right)^k (1 - t^2)^{k+(d-3)/2}.$$

The polynomials $P_k$ are orthogonal in $L^2([-1,1], d\nu)$ where the measure $d\nu$ is given by $d\nu(t) = (1 - t^2)^{(p-3)/2} dt$, and we have

$$\int_{-1}^{1} P_k^2(t)(1 - t^2)^{(p-3)/2} dt = \frac{\omega_{p-1}}{\omega_{p-2}} \frac{1}{N(p,k)}, \tag{17}$$

where $\omega_{d-1} = \frac{2\pi^{d/2}}{\Gamma(d/2)}$ denotes the surface of the sphere $\mathbb{S}^{d-1}$ in $d$ dimensions. Using the addition formula (16) and orthogonality of spherical harmonics, we can show

$$\int P_j(\langle w, x \rangle) P_k(\langle w, y \rangle) d\tau(w) = \frac{\delta_{jk}}{N(p,k)} P_k(\langle x, y \rangle) \tag{18}$$

Further, we have the recurrence relation [22, Eq. 4.36]

$$t P_k(t) = \frac{k}{2k+p-2} P_{k-1}(t) + \frac{k+p-2}{2k+p-2} P_{k+1}(t), \tag{19}$$

for $k \geq 1$, and for $k = 0$ we simply have $t P_0(t) = P_1(t)$.

The Funk-Hecke formula is helpful for computing Fourier coefficients in the basis of spherical harmonics in terms of Legendre polynomials: for any $j = 1, \ldots, N(p,k)$, we have

$$\int f(\langle x, y \rangle) Y_{k,j}(y) d\tau(y) = \frac{\omega_{p-2}}{\omega_{p-1}} Y_{k,j}(x) \int_{-1}^{1} f(t) P_k(t)(1 - t^2)^{(p-3)/2} dt. \tag{20}$$

## C.2 Dot-product kernels and spherical harmonics

In this section, we provide some background on how spherical harmonics can be used for obtaining descriptions of the RKHS for dot-product kernels on the $p-1$ sphere [46, 47]. We then recall results from Bach [6] on such a description for kernels arising from positively-homogeneous activations (*i.e.*, arc-cosine kernels), and on how differentiability of functions on the sphere relates to the RKHS.

For a positive-definite kernel $K(x,y) = \kappa(\langle x,y \rangle)$ defined on the $p-1$ sphere $\mathbb{S}^{p-1}$, we have the following Mercer decomposition:

$$\kappa(\langle x,y \rangle) = \sum_{k=0}^{\infty} \mu_k \sum_{j=1}^{N(p,k)} Y_{k,j}(x)Y_{k,j}(y) = \sum_{k=0}^{\infty} \mu_k N(p,k) P_k(\langle x,y \rangle), \tag{21}$$

where $\mu_k$ is obtained by computing Fourier coefficients of $\kappa(\langle x,\cdot \rangle)$ using (20):

$$\mu_k = \frac{\omega_{p-2}}{\omega_{p-1}} \int_{-1}^{1} \kappa(t) P_k(t)(1-t^2)^{(p-3)/2} dt. \tag{22}$$

Note that we must have $\mu_k \geq 0$ for all $k$, since positive definiteness of $k$ would be violated if this were not true. Then, the RKHS $\mathcal{H}$ consists of functions of the form

$$f(x) = \sum_{k=0}^{\infty} \sum_{j=1}^{N(p,k)} a_{k,j} Y_{k,j}(x), \tag{23}$$

$$\text{s.t.} \quad \|f\|_{\mathcal{H}}^2 = \sum_{k=0}^{\infty} \sum_{j=1}^{N(p,k)} \frac{a_{k,j}^2}{\mu_k} < \infty. \tag{24}$$

In particular, this requires that $a_{k,j} = 0$ for all $j$, for any $k$ such that $\mu_k = 0$.

**Relationship with differentiability.** Following Bach [6, Appendix D.3], if $f$ is $s$-times differentiable with derivatives bounded by $\eta$, then we have $\|(-\Delta)^{s/2}f\|_{L^2(\mathbb{S}^{p-1})} \leq \eta$, where $\Delta$ is the Laplace-Beltrami operator on the sphere [5]. Given a function $f$ as in (23), we can use the fact that $Y_{k,j}$ is an eigenfunction of $-\Delta$ with eigenvalue $k(k+p-2)$ (see [5, 22]) and write $a_{k,j} = a'_{k,j}/(k(k+p-2))^{s/2}$ for $k \geq 1$, where $a'_{k,j}$ are the Fourier coefficients of $(-\Delta)^{s/2}f$ and satisfy $\sum_{k,j} a'^2_{k,j} \leq \eta^2$. We then have

**Lemma 16.** *Assume $f$ takes the form* (23)*, with $a_{k,j} = 0$ for all $j$ when $\mu_k = 0$, and that $f$ is $s$-times differentiable with derivatives bounded by $\eta$. Then, if $\max_{k \geq 1, \mu_k \neq 0} 1/(k^{2s}\mu_k) < C$, we have $f \in \mathcal{H}$ with $\|f\|_{\mathcal{H}}^2 \leq C'\eta^2$, with $C' = 1/\mu_0 + C$ if $\mu_0 \neq 0$, or $C' = C$ if $\mu_0 = 0$.*

Indeed, under the stated conditions, we have

$$\|f\|_{\mathcal{H}}^2 = \sum_{k=0}^{\infty} \sum_{j=1}^{N(p,k)} \frac{a_{k,j}^2}{\mu_k} \leq \frac{a_{0,1}^2}{\mu_0} + \max_{k \geq 1, \mu_k \neq 0} 1/(k^{2s}\mu_k) \sum_{k,j} a'^2_{k,j}$$

$$\leq \frac{1}{\mu_0}\eta^2 + C\|(-\Delta)^{s/2}f\|_{L^2(\mathbb{S})}^2 \leq C'\eta^2.$$

**Values of $\mu_k$ for arc-cosine 0/1 kernels [6].** We recall values of the eigenvalues $\mu_k$ for arc-cosine kernels of degree 0 and 1, which are obtained in [6]. For any non-negative integer $\alpha \geq 0$, Bach [6] considers the positively homogeneous activation $\sigma_\alpha(u) = (u)_+^\alpha$ and derives the following quantities for $k \geq 0$:

$$\lambda_{\alpha,k} = \frac{\omega_{p-2}}{\omega_{p-1}} \int_{-1}^{1} \sigma_\alpha(t) P_k(t)(1-t^2)^{(p-3)/2} dt.$$

This can be used to derive the decompositions of the arc-cosine kernels introduced in Section 2, which are defined using expectations on Gaussian variables, but can be expressed using expectations on the sphere as follows:

$$\kappa_\alpha(\langle x,y \rangle) = 2\,\mathbb{E}_{w \sim \mathcal{N}(0,1)}[\sigma_\alpha(\langle w,x \rangle)\sigma_\alpha(\langle w,y \rangle)]$$

$$= 2\,\mathbb{E}_{w \sim \mathcal{N}(0,1)}[\|w\|^{2\alpha}\sigma_\alpha(\langle w/\|w\|,x \rangle)\sigma_\alpha(\langle w/\|w\|,y \rangle)]$$

$$= 2\,\mathbb{E}_{w \sim \mathcal{N}(0,1)}[\|w\|^{2\alpha}]\int \sigma_\alpha(\langle w,x \rangle)\sigma_\alpha(\langle w,y \rangle)d\tau(w),$$

where we used $\alpha$-homogeneity of $\sigma_\alpha$ and the rotational symmetry of the normal distribution, which implies that $w/\|w\|$ is uniformly distributed on the sphere, and independent from $\|w\|$.

For a fixed $w$, we can express

$$\sigma_\alpha(\langle w, x \rangle) = \sum_{k=0}^{\infty} \lambda_{\alpha,k} N(p,k) P_k(\langle w, y \rangle).$$

Then, using (18), we have

$$\kappa_\alpha(\langle x, y \rangle) = 2\, \mathbb{E}_{w \sim \mathcal{N}(0,1)}[\|w\|^{2\alpha}] \sum_{k=0}^{\infty} \lambda_{\alpha,k}^2 N(p,k) P_k(\langle x, y \rangle).$$

For $\alpha = 0, 1$, this yields decompositions (21) of $\kappa_0, \kappa_1$ with $\mu_{0,k} = 2\lambda_{0,k}^2$ and $\mu_{1,k} = 2p\lambda_{1,k}^2$. Bach [6] then shows the following result on the decomposition of $\sigma_\alpha$, which we translate to decompositions of kernel functions $\kappa_\alpha$.

**Lemma 17** (Decomposition of $\sigma_\alpha$ and $\kappa_\alpha$ [6]). *For the activation $\sigma_\alpha$ on the $p-1$ sphere, we have*

- $\lambda_{\alpha,k} \neq 0$ *if $k \leq \alpha$;*

- $\lambda_{\alpha,k} = 0$ *if $k > \alpha$ if $k = \alpha \mod 2$;*

- $|\lambda_{\alpha,k}| \sim C_\lambda(p,\alpha) k^{-p/2-\alpha}$ *otherwise, for some constant $C_\lambda(p,\alpha)$ depending on $p$ and $\alpha$.*

*For $\alpha \in \{0,1\}$, the eigenvalues for the corresponding kernel $\kappa_\alpha$ then satisfy*

- $\mu_{\alpha,k} > 0$ *if $k \leq \alpha$;*

- $\mu_{\alpha,k} = 0$ *if $k > \alpha$ if $k = \alpha \mod 2$;*

- $\mu_{\alpha,k} \sim C_\mu(p,\alpha) k^{-p-2\alpha}$ *otherwise, with $C_\mu(p,\alpha) = 2p^\alpha C_\lambda(p,\alpha)^2$.*

Note that the zero eigenvalues imply that a function $f$ of the form (23) must have $a_{k,j} = 0$ for $k > \alpha$ and $k = \alpha \mod 2$ in order to be in the RKHS for $\kappa_\alpha$. A sufficient condition for this to hold is that $f$ is even (resp. odd) when $\alpha$ is odd (resp. even) [6].

### C.3 Decomposition of the NTK for two-layer ReLU networks

Recall that the NTK for the two-layer ReLU network with inputs on the sphere is given by

$$\kappa(u) = u\kappa_0(u) + \kappa_1(u).$$

We now prove Proposition 5, which shows that the Mercer decomposition in spherical harmonics (21) for this NTK satisfies:

- $\mu_0, \mu_1 > 0$
- $\mu_k = 0$ if $k = 2j+1, j \geq 1$
- $\mu_k \sim C(p)k^{-p}$ otherwise, with $C(p) = C_\mu(p,0)$.

*Proof of Proposition 5.* Using (22) and the recurrence relation (19) for Legendre polynomials, as well as $tP_0(t) = P_1(t)$, we have

$$\mu_0 = \mu_{0,1} + \mu_{1,0}$$
$$\mu_k = \frac{k}{2k+p-2}\mu_{0,k-1} + \frac{k+p-2}{2k+p-2}\mu_{0,k+1} + \mu_{1,k}, \quad \text{for } k \geq 1.$$

By Lemma 17, we have the desired properties. □

We now briefly discuss how to adapt the approximation results of Bach [6] to our setting.

*Proof sketch of Corollaries 6 and 7.* In Appendix D of [6], Bach defines candidate functions $g : \mathbb{S}^{p-1}$ from functions $p \in L^2(\mathbb{S}^{p-1})$ as $g(x) = Tp(x) := \int p(w)\sigma_\alpha(w^\top x)d\tau(w)$, with RKHS norm (denoted $\gamma_2(g)$ in [6]) given by the smallest $\|p\|_{L^2}$ for $p$ such that $g = Tp$. In our case with the NTK, we may simply consider the operator $\Sigma^{1/2}$ instead (the self-adjoint square root of the integral operator $\Sigma$ of $\kappa$, using notations from [7]; see also [18]), which simply multiplies each fourier coefficient $a_{k,j}$ in decomposition (23) by $\sqrt{\mu_k}$, and obeys the required properties (in fact, $T$ and $\Sigma^{1/2}$ are two different square roots of $\Sigma$ [7]).

The proofs can then be adapted directly, by noticing that $\sqrt{\mu_k}$ has the same decay properties as $\lambda_{\alpha,k}$ with $\alpha = 0$. For Corollary 6, we also provide the key proof ingredients in our framework in Lemma 16 for completeness. $\qquad\square$

The proof for the homogeneous RKHS is given below.

*Proof of Proposition 8.* The kernel $K$ can be written as

$$K(x, x') = \langle \|x\|\Phi\left(\frac{x}{\|x\|}\right), \|x'\|\Phi\left(\frac{x'}{\|x'\|}\right)\rangle_{\mathcal{H}},$$

where $\Phi(\cdot)$ is the kernel mapping of the kernel $\kappa$ on the sphere. Then the RKHS $\bar{\mathcal{H}}$ can be characterized by the following classical result (see, *e.g.*, [44, §2.1] or [6, Appendix A]):

$$\bar{\mathcal{H}} = \underbrace{\{x \mapsto \langle g, \|x\|\Phi\left(\frac{x}{\|x\|}\right)\rangle_{\mathcal{H}} : g \in \mathcal{H}\}}_{=:f_g}$$
$$\|f_g\|_{\bar{\mathcal{H}}} = \inf\{\|g'\|_{\mathcal{H}} : g' \in \mathcal{H} \text{ s.t. } f_g = f_{g'}\}.$$

Note that the condition $f_g = f_{g'}$ implies in particular that $f_g$ and $f_{g'}$ are equal on the sphere, and thus that $g = g'$, so that the infimum is simply equal to $\|g\|_{\mathcal{H}}$. This concludes the proof.

$\qquad\square$

# D   Details on Numerical Experiments

This section provides more details on the experimental setup used for obtaining Figure 1, which closely follows the setup in [11, Section 3.4].

We consider images of handwritten digits from the Infinite MNIST dataset [32], which consists of 28x28 grayscale MNIST digits augmented with small translations and deformations. Translations are chosen at random from one of eight possible directions, while deformations are generated by considering small smooth deformations $\tau$, and approximating $L_\tau x$ using a tangent vector field $\nabla x$ containing partial derivatives of the signal $x$ along the horizontal and vertical image directions. We introduce a deformation parameter $\alpha$ to control deformation size. The images are then given by

$$L_{\alpha\tau}x(u) = x(u - \alpha\tau(u)) \approx x(u) - \alpha\tau(u) \cdot \nabla x(u).$$

Figure 2 shows examples of different deformations, with various values of $\alpha$, with or without translations, generated from a reference image of the digit "5". In addition, one may consider that a given reference image of a handwritten digit can be deformed into different images of the same digit, and perhaps even into a different digit (*e.g.*, a "1" may be deformed into a "7"). Intuitively, the latter transformation corresponds to a "larger" deformation than the former, so that a prediction function that is stable to deformations should be preferable for a classification task.

The architecture underlying the kernels considered in Figure 1 consists of 2 convolutional layers with patches of size 3x3, followed by ReLU activations, and Gaussian pooling layers adapted to subsampling factors 2 for the first layer and 5 for the second. Patch extraction is performed with zero padding in order to preserve the size of the previous feature map. For a subsampling factor $s$, we apply a normalized Gaussian filter with scale $\sigma = s/\sqrt{2}$ and size $(2s + 1) \times (2s + 1)$, before downsampling. Our C++ implementation for computing the full kernel given two images is based on dynamic programming and is available at `https://github.com/albietz/ckn_kernel`.

Figure 2: MNIST digits with transformations considered in our numerical study of stability. Each row gives examples of images from a set of digits that are compared to a reference image of a "5". From top to bottom: deformations with $\alpha = 3$; translations and deformations with $\alpha = 1$; digits from the training set with the same label "5" as the reference digit; digits from the training set with any label.

The plots in Figure 1 show average relative distance in the RKHS between a reference image and images from various sets of 20 images (either generated transformations, or images appearing in the training set). The figures consider deformations of varying size $\alpha \in \{0.01, 0.03, 0.1, 0.3, 1, 3\}$. For a given kernel $K$ and set $S$ of images, the average relative distance to an image $x$ is given by

$$\frac{1}{|S|} \sum_{x' \in S} \frac{\|\Phi(x') - \Phi(x)\|_{\mathcal{H}}}{\|\Phi(x)\|_{\mathcal{H}}} = \frac{1}{|S|} \sum_{x' \in S} \frac{\sqrt{K(x, x) + K(x', x') - 2K(x, x')}}{\sqrt{K(x, x)}},$$

where $\mathcal{H}$ denotes the RKHS associated to $K$ and $\Phi$ the kernel mapping. We normalize by $\|\Phi(x)\|_{\mathcal{H}}$ in order to reduce sensitivity to the choice of kernel.