[Reviews · NeurIPS 2019]

Reviewer 1



Study of the Neural Tangent Kernel (Jacot et al. 2018) is a compelling approach to understanding the dynamics of learning in neural networks. As acknowledged by the authors, while this kernel determines the dynamics only for very wide networks, better understanding of this simplified regime could be a first step towards a fuller understanding of learning dynamics more generally. In this work, the authors study the smoothness, approximation and stability of this kernel for shallow fully connected and convolutional networks. It has been recognized for some time that since overparametrized neural networks do not overfit, the choice of architecture and/or optimization algorithm must bias the solutions found towards ones that generalize well. In the past, it has been difficult to understand this implicit bias except in very simple architectures, and studying it in the wide network regime is another step towards a better understanding of this phenomenon. The presentation is very clear for the most part, and the authors make an effort to distinguish their contributions from previous work. They are also honest about the limitations of the setting of the analysis (the only exception is perhaps the issue of the decomposition in terms of spherical harmonics, see below). When presenting the Mercer decomposition in terms of spherical harmonics in Proposition 5, isn't one assuming a uniform data distribution over the sphere? If this were not the case, then the RKHS in eq. 11 would take a different form (since one could choose f to be supported on the entire sphere). If this assumption is made elsewhere in the text it might be good to state it clearly in this section. It is also useful to note that in general one would expect realistic data distributions to be supported on low dimensional subsets of the sphere. This limits the applicability of the results in this section. Given the known stability results for convolutional networks, is it surprising that the convolutional NTK feature map is stable as well?

Reviewer 2



Post author response: After reading other reviews and author's response, my evaluation still holds. I thank the authors for the thoughtful response and will be looking forward to see future developments in this direction. ______________________________________ In the infinite width limit, neural network’s gradient flow training dynamics is well captured by Neural Tangent Kernel. The advantage of this picture of understanding neural network is providing angle to study kernel properties to understand deep neural networks which has been a hard challenge theoretically. Although applicability of “lazy training” via NTK to realistic model / dataset is still not clear, it is an important theoretical landmark to have a good understanding of this kernel capturing neural network training in a certain limit. The authors set out to study inductive bias of Neural Tangent Kernel which is very timely and important contribution. Authors show that NTK with two-layer ReLU networks are 1) not Lipshitz smooth but satisfy weaker Holder smoothness property. 2) Also studying CNN NTKs stability shows less stability compared to kernels obtained by fixing weight of all layers except the last layer. 3) Using spherical harmonics decomposition show that eigenvalues decay slower than ArcCos kernel that would correspond to ReLU network kernel with fixed weights except for the last layer. Interesting observation pointed out by the authors is that the finding show tradeoff between stability and approximation where better approximation property captured by NTK is tradedoff by less stable/smooth property. While few recent concurrent work discuss NTK for convolutional networks, the current submission also provide definition of NTK for CNN, independently, also generalizing to linear operators that has not been considered in other works such as patch extraction and pooling operators. Few obvious limitation is that the analysis especially looking at the spherical harmonic decomposition assumes that input data is uniform sample from hyper-sphere. For real dataset, it is unclear how kernel spectral properties would be similar or different from simple toy data domain. While all of the work is studying property of kernels theoretically, it is a weakness of the work not showing any empirical support of the inductive bias described in the paper. Nit: Line 167 “NTK kernel” repeats `kernel’ twice. For references to GP limit of neural networks, one should also cite [1] along with papers already been cited. [1] Alexander Matthews et al., Gaussian Process Behaviour in Wide Deep Neural Networks, ICLR 2018

Reviewer 3



I have read the response. ---------------------------------------- This paper makes strong contributions to the recent line of work that demonstrates the equivalence between over-parameterized neural networks and a new class of kernels, neural tangent kernels (NTK). The new class of kernels is different from classical kernels like Gaussian kernel and we do not have a good understanding on why it gives better performance than classical kernels. This paper initiates a rigorous study on the properties of NTK. This paper provides a smoothness analysis of NTK induced by two-layer neural network. The result may be useful for understanding the robustness of neural network as well. The paper further provides approximation analysis of the NTK. I really like the result as it demonstrates the advantage of NTK comparing with previous kernels. The analyses on the smoothness and stability of CNTK are also interesting and authors did a good job in establishing the connection with previous work on CNNs in this direction. Overall I really like this paper and recommend to accept. Minor: 1. If I understand correctly, the proof for deriving CNTK relies on the sequential limit. This should be stated explicitly in the final version.

[Author Response · NeurIPS 2019]

We thank all reviewers for their positive reviews and valuable comments. We begin by addressing questions that are of general concern among multiple reviewers, and later respond to questions individually to each reviewer.

**Role of the data distribution.** Reviewers 1 and 2 raise the interesting question of the data distribution and its effect on the spectral properties of the kernel. However, note that our study of the Mercer decomposition is aimed at providing a clear description of the functions in the RKHS, their norm, as well as their approximation properties, *irrespectively* of the specific data distribution. In particular, one can obtain a Mercer decomposition for any measure, and the approximation properties given in Corollary 6 and 7 rely specifically on the spherical harmonic decomposition with a uniform distribution on the sphere.

While our analysis does not provide a precise study of estimation error for specific data distributions, it does yield insight for the (somewhat crude) uniform bounds based on Rademacher complexity, which only depend on the data through the norm of the learned function $\hat{f}$ (and the radius $R$), taking the form $O(\|\hat{f}\|R/\sqrt{n})$. In the context of NTK, the papers [1, 2] derive such bounds where $\hat{f}$ is the minimum-norm interpolating solution. For more refined bounds based on eigenvalues of the integral operator, one would then require a spectral decomposition of the kernel w.r.t. the data distribution, which is more difficult to obtain (though the eigenvalue decay may be preserved, e.g., if the data distribution is absolutely continuous w.r.t. the uniform distribution on the sphere). We will be happy to clarify this further in the paper.

**Role of depth.** We agree that a limitation of the paper is that the approximation results of Section 3.2 are limited to two-layer fully-connected networks. The extension to a two-layer CNN with global average pooling is straightforward, with an eigenvalue decay similar to the fully-connected case but which only depends on the dimension of a patch rather than the full signal. The study of approximation for deeper networks is more complicated and is left for future work. We will state this more explicitly in the paper. The smoothness and stability results do apply to deep CNNs, and in particular depth is important for deformation stability, since the bound in Proposition 12 improves with smaller patches (i.e., small $\beta$): indeed, with appropriate pooling and downsampling, a deeper architecture is needed in order to reach a fixed target level of translation invariance with small patches at each layer [8]. We will clarify this further in the paper.

**Empirical validation.** We conducted numerical experiments in order to assess stability and approximation properties, comparing the NTK to the simpler kernel with all layers fixed but the last, for a three-layer convolutional architecture on MNIST digits. Considering deformations from the "infinite MNIST" dataset, we indeed observe that the stability of the NTK kernel mapping is weaker, with a faster growth as a function of the deformation size for small deformations. Regarding approximation, we computed interpolating solutions $\hat{f}$ on binary classification problems with a dozen digits in each class for the two kernels, and found that the quantity $\|\hat{f}\|_{\mathcal{H}}R$ (where $R$ is the average of norms $\|\Phi(x_i)\|_{\mathcal{H}}$, for normalization purposes) is always smaller for the NTK, suggesting it indeed has better approximation properties. We will be happy to include these results in the paper, if it is accepted.

**R1.** We thank the reviewer for his positive comments. The questions related to data distribution and depth are addressed above.
- "... is it surprising ...": given an appropriate CNN architecture, stability is indeed not surprising, but we find it interesting to characterize stability for the NTK, contrast it with approximation results, and compare it with known deep convolutional kernels.
- "... lower bound on stability ...": this is an interesting point, which is partly discussed in earlier papers on deformation stability (e.g. Section 3.2 in [8]), though without a precise lower bound. One way to see this instability is by constructing a function $f$ in the RKHS based on a large filter with very high frequencies, but with norm less than one. This yields a lower bound on $\|\Phi(x) - \Phi(x')\| \geq f(x) - f(x')$ which can be made arbitrarily unstable due to high frequencies, even when $x'$ is a small deformation of $x$.

**R2.** We thank the reviewer for pointing out the Matthews reference, which we will include in the paper. The concerns on the data distribution, depth and empirics are addressed above.

**R3.** We thank the reviewer for his remarks. We will state the requirement on sequential limit more explicitly in the paper. While our stability bounds partially illustrate the benefits of depth and convolution, we agree that extending the approximation results to deep CNNs would be interesting (see above).

# References

[1] S. Arora, S. S. Du, W. Hu, Z. Li, and R. Wang. Fine-grained analysis of optimization and generalization for overparameterized two-layer neural networks. In *ICML*, 2019.

[2] Y. Cao and Q. Gu. Generalization bounds of stochastic gradient descent for wide and deep neural networks. *preprint arXiv:1905.13210*, 2019.


[Meta-Review · NeurIPS 2019]

This paper studies the inductive bias of neural tangent kernels (NTKs) which is important in understanding the dynamics of learning in neural networks. There is a strong consensus among the reviewers that this paper provides solid theoretical results which help advance the theoretical understanding of deep neural networks. I also think that the topic addressed in this paper would be of interest to people in the NeurIPS community and would help bridge the gap between deep learning and theory of kernel methods. Hence, I would recommend to accept the paper.